# PROVABLE DYNAMIC REGULARIZATION CALIBRATION

## ABSTRACT

Miscalibration in deep learning refers to the confidence of the model does not match the performance. This problem usually arises due to the overfitting issue in deep learning models, resulting in overly confident predictions during testing. Existing methods typically prevent overfitting and mitigate miscalibration by adding a maximum-entropy regularizer to the objective function. The objective of these method can be understood as seeking a model that not only fits the ground-truth labels by *increasing the confidence* but also maximizes the entropy of predicted probabilities by *decreasing the confidence*. However, previous methods cannot provide clear guidance on when to increase the confidence (known knowns) or decrease the confidence (known unknowns), leading to the two conflicting optimization objectives (increasing but also decreasing confidence). In this work, we propose a simple yet effective method called dynamic regularization calibration (DRC), to address this trade-off by exploring outlier samples within the training set, resulting in a reliable model that can admit it knows somethings and does not know others. DRC effectively fits the labels for in-distribution samples while applying regularization to potential outliers dynamically, thereby obtaining robust calibrated model. Both theoretical and empirical analyses demonstrate the superiority of DRC compared with previous methods.

## 1 INTRODUCTION

Deep neural networks have achieved remarkable progress on various tasks (LeCun et al., 2015). However, current deep learning classification models often suffer from poor calibration, i.e., their confidence scores fail to accurately reflect the accuracy of the classifier, especially exhibiting excessive overconfidence (Guo et al., 2017). This leads to challenges when deploying them to safety-critical downstream applications like autonomous driving (Bojarski et al., 2016) or medical diagnosis (Esteva et al., 2017), since the decisions cannot be trusted even with high confidence. Calibrating the confidence of deep classifiers is crucial for the successful deployment of deep neural networks.

Various regularization-based methods are proposed to calibrate the deep classifiers. Specifically, existing empirical evidences reveal that weight decay and label smoothing can improve calibration performance (Guo et al., 2017; Müller et al., 2019). Moreover, most previous methods directly modify the objective function by implicitly or explicitly adding a maximum-entropy regularizer during training, such as penalizing confidence (Pereyra et al., 2017), focal loss (Mukhoti et al., 2020; Wang et al., 2021; Ghosh et al., 2022; Tao et al., 2023), and logit normalization (Wei et al., 2022). The underlying objective of these methods can be intuitively understood as minimizing the classification loss by *increasing confidence* corresponding to the ground-truth label, while simultaneously maximizing the predictive entropy of the predicted probability by *reducing confidence*. In short, previous methods strive to adjust the confidence while accurately classifying to avoid miscalibration.

However, existing methods lack clear guidance on how to adjust confidence, which may lead to unreliable prediction when training in practical scenarios where the training set contains both easy samples and challenging samples (e.g., outliers). On the one hand, existing methods attempt to apply regularization to prevent the model from overly confident, which may lead to lower confidence even for easy samples that should be classified accurately with high confidence. On the other hand, they still strive to reduce the classification loss by increasing the confidence corresponding to the ground-truth label to fit their labels, which could lead to higher confidence for challenging samples that are

Figure 1: The motivation of the DRC. Previous regularization-based methods aim for accurate classification while maximizing predictive entropy, resulting in conflicting optimization goals of simultaneously increasing and decreasing confidence. To address this issue, DRC dynamically applies regularization to avoid these conflicting optimization objectives, assigning higher confidence to simple samples and lower confidence to challenging samples.

difficult to classify accurately. Therefore, when the training set contains both easy and challenging samples, the potential limitations of existing methods are exposed. Unfortunately, such scenarios are prevalent, arising from various reasons such as differences in inherent sample difficulty (Seedat et al., 2022; Lorena et al., 2019; Vasudevan et al., 2022; Toneva et al., 2018), data augmentation (Yun et al., 2019; Cubuk et al., 2020) or the existence of multiple subgroups in the data (Yao et al., 2022; Han et al., 2022; Yang et al., 2023).

In summary, the lack of clear supervision on how to adjust confidence may lead to three significant problems of existing methods: (1) They may overly regularize the predictions of model, resulting in higher predicted entropy even for easy samples, potentially leading to an *under-confident* model (Wang et al., 2021); (2) They enforce the model to *over-confidently* classify all training samples, even though that may be potentially unclassifiable outliers; (3) Striving for accurate classification on the training samples while simultaneously maximizing prediction entropy represents opposing goals, making it *difficult in balancing* between the two objectives.

To this end, as shown in Fig. 1, we propose a simple method (DRC) to solve the above three problems by implicitly constructing a probability of whether a sample should be known to the model and then impose dynamic regularization to different samples. Specifically, we avoid imposing regularization to increase predicted entropy for easy samples, which prevents undermining model confidence for easy samples (known knows - known what can be confidently classified). Meanwhile, DRC prevents deep learning models from miscalibration by increasing the predicted entropy on potential outlier samples (known unknowns - known what can not be classified confidently). In this way, DRC can elegantly balance the two opposing goals of accurately classifying training samples and maximizing the entropy of predicted probabilities. By actively distinguishing known and unknown samples, DRC can achieve more robust calibration. Both theoretical and experimental results demonstrate the effectiveness of DRC. The contributions of this paper are as follows:

- We propose a simple yet effective method that improves previous regularization-based calibration approaches by informing the model what it should and should not know, avoiding problems caused by the lack of explicit confidence guidance in previous works.

- We provide theoretical analysis that proves the superiority of DRC in achieving lower calibration error compared to previous regularization-based methods.

- We conduct extensive experiments on various settings and datasets. The experimental results, with DRC achieving the best performance (87 times out of 108 experiments), strongly suggest DRC outperforms previous approaches in calibration.

## 2 RELATED WORK

**Confidence calibration**. A well-calibrated classifier can be approached through two main methods: post-hoc calibration and regularization-based calibration. In post-hoc calibration, after the classifier is trained, its predicted confidence is adjusted by training extra parameters on a validation set to improve calibration without modifying the original model (Guo et al., 2017; Yu et al., 2022). An example is temperature scaling (Guo et al., 2017), where a temperature parameter is trained on the validation set to scale the predicted probability distribution. On the other hand, regularization-based calibration avoids classifier miscalibration by incorporating regularization techniques during

the training of deep neural networks. This includes strategies such as training with strong weight decay (Guo et al., 2017), label smoothing (Müller et al., 2019), penalizing confidence (Pereyra et al., 2017), focusing on under-confident samples (Mukhoti et al., 2020; Ghosh et al., 2022; Wang et al., 2021; Tao et al., 2023), and constraining the norm of logits (Wei et al., 2022; 2023).

**Uncertainty estimation**. The primary goal of uncertainty estimation is to obtain the reliability of the model. Traditional uncertainty estimation methods utilize ensemble learning (Lakshminarayanan et al., 2017; Liu et al., 2019) and Bayesian neural networks (Neal, 2012; MacKay, 1992; Denker & LeCun, 1990; Kendall & Gal, 2017) to obtain the distribution of predictions, from which uncertainty can be estimated using the variance or entropy of the distribution. Recently, several regularization-based uncertainty estimation methods have been proposed. These methods typically obtain uncertainty by applying regularization to neural networks using either additional out-of-distribution dataset (Malinin & Gales, 2018) or training set samples (Sensoy et al., 2018).

**Out-of-distribution (OOD) detection**. OOD detection aims to distinguish potential OOD samples to avoid unreliable predictions. One of the primary approaches is to modify the loss function, thereby constraining the model to effectively identify potential OOD samples. Specifically, existing methods usually impose constraints on the model, such as having a uniform prediction distribution (Lee et al., 2018; Hendrycks et al., 2018; Choi et al., 2023) or higher energy (Katz-Samuels et al., 2022; Liu et al., 2020; Du et al., 2021; Chen et al., 2021), particularly on the additional outlier training set.

**Comparison with existing works**. The core principle shared by regularization-based approaches in above topics is to impose regularization on neural networks during training to prevent overfitting. These methods can be roughly categorized into two types: (1) *training without outlier data* – These methods aim to simultaneously achieve accurate classification and regularize the model to maximize prediction entropy on the training data. However, as shown in the introduction, there is an underlying trade-off between achieving high accuracy and high entropy, making it challenging to balance these two objectives. (2) *training utilizing additional outlier data* – Methods along this line leverage external outlier datasets to regularize the model. The prohibitively large sample space of outliers requires additional sample selection strategies to obtain outliers that can effectively regularize the behavior of the model (Ming et al., 2022). However, the artificially designed sample selection strategies may introduce biases into the distribution of outlier samples, resulting in a failure to characterize the true distribution of OOD data encountered at test time. Besides, how to utilize outlier samples to improve calibration is still an open problem. In contrast to the aforementioned methods, DRC explores and leverages information from naturally occurring outlier samples within the training set that have a high probability of being similar to ones encountered at test time. In this way, DRC eliminates the need for additional external outlier datasets and the corresponding sample selection strategies, while avoiding the trade-off between model performance and regularization.

## 3 DYNAMIC REGURLARIZATION CALIBRATION

**Notations**. Let $\mathcal{X}$ and $\mathcal{Y} = \{1, 2, \cdots, K\}$ be the input and label space respectively, where $K$ denotes the number of classes. The training dataset $\mathcal{D} = \{\boldsymbol{x}_i, y_i\}_{i=1}^n$ consists of $n$ samples independently drawn from a training distribution $P$ over $\mathcal{X} \times \mathcal{Y}$. The label $y$ can also be represented as a one-hot vector $\boldsymbol{y} = [y^1, \cdots, y^K]$, where $y^k = \mathbb{1}(k = y)$ and $\mathbb{1}(\cdot)$ is the indicator function. The goal of the classification task is to train a model $f_{\boldsymbol{\theta}} : \mathcal{X} \to \Delta^K$ parameterized with $\boldsymbol{\theta} \in \Theta$, where $\Delta^K$ denotes the classification probability space. This model can produce a predictive probability vector $f_{\boldsymbol{\theta}}(\boldsymbol{x}) = [f_{\boldsymbol{\theta}}^1(\boldsymbol{x}), \cdots, f_{\boldsymbol{\theta}}^K(\boldsymbol{x})]$ for input $\boldsymbol{x}$. The predicted label and the corresponding confidence score can be obtained with $\hat{y} := \arg\max f_{\boldsymbol{\theta}}(\boldsymbol{x})$ and $\hat{\mathbb{P}}(\boldsymbol{x}) := \max f_{\boldsymbol{\theta}}(\boldsymbol{x})$, respectively. Moreover, $\hat{\mathbb{P}}(\boldsymbol{x})$ is generally considered as the estimated probability that $\hat{y}$ is the correct label for $\boldsymbol{x}$.

**Confidence calibration**. A model is considered perfectly calibrated (Guo et al., 2017) when the estimated probability or confidence score $\hat{\mathbb{P}}(\boldsymbol{x})$ precisely matches the probability of the model classifying $\boldsymbol{x}$ correctly, i.e., $\mathbb{P}(\hat{y} = y \mid \hat{\mathbb{P}}(\boldsymbol{x}) = p) = p$ for all $p \in [0, 1]$. For instance, for samples that obtain an estimated confidence of 0.8, a perfectly calibrated neural network should have an accuracy of 80%. However, recent empirical observations have revealed that widely used deep neural networks often exhibit overconfidence due to overfitting on the training data (Guo et al., 2017; Mukhoti et al., 2020). The objective of confidence calibration is to obtain a model whose predicted confidence scores can accurately reflect the predictive accuracy of the model.

## 3.1 Regularization in Neural Networks for Calibration

This paper focuses on applying regularization to prevent overfitting, thus alleviating overconfidence. Existing regularization-based methods typically incorporate implicit or explicit regularization on the model by modifying the objective function. We first present two representative methods (more methods detailed in the appendix (Sec. A)) and then show that they share similar goals and limitations.

**Label smoothing** is a regularization technique that involves training the neural network with softened target labels $\widetilde{\boldsymbol{y}} = [\widetilde{y}^1, \cdots, \widetilde{y}^K]$ to improve the reliability of confidence (Müller et al., 2019), where $\widetilde{y}^k = (1 - \epsilon)y^k + \epsilon/K$ and $0 < \epsilon < 1$ controls the strength of smoothing. When using cross-entropy loss, the label smoothing loss can be decomposed as follows:

$$\mathcal{L}_{ce}(f_{\boldsymbol{\theta}}(\boldsymbol{x}), \widetilde{\boldsymbol{y}}) = (1 - \epsilon)\mathcal{L}_{ce}(f_{\boldsymbol{\theta}}(\boldsymbol{x}), \boldsymbol{y}) + \epsilon\mathcal{L}_{ce}(f_{\boldsymbol{\theta}}(\boldsymbol{x}), \boldsymbol{p}_u), \tag{1}$$

where $\mathcal{L}_{ce}$ denotes the cross-entropy loss and $\boldsymbol{p}_u = [1/K, \cdots, 1/K]$ is a uniform distribution.

**Focal loss** and other related calibration methods implicitly regularize the deep neural network by increasing the weight of samples with larger losses. This prevents the model from overfitting due to excessive attention to simple samples (Mukhoti et al., 2020; Ghosh et al., 2022; Tao et al., 2023). The classical focal loss is defined as $\mathcal{L}_f(f_{\boldsymbol{\theta}}(\boldsymbol{x}), \boldsymbol{y}) = -(1 - f_{\theta}^y(\boldsymbol{x}))^\gamma \log f_{\theta}^y(\boldsymbol{x})$, where $\gamma$ is a hyperparameter that controls the reweighting strength, allowing higher weight to be assigned to samples with lower confidence in the correct label. A general form of focal loss can be formulated as an upper bound of the regularized cross-entropy loss (Mukhoti et al., 2020),

$$\mathcal{L}_f(f_{\boldsymbol{\theta}}(\boldsymbol{x}), \boldsymbol{y}) \geq \mathcal{L}_{ce}(f_{\boldsymbol{\theta}}(\boldsymbol{x}), \boldsymbol{y}) - \gamma\mathcal{H}(f_{\theta}(\boldsymbol{x})), \tag{2}$$

where $\mathcal{H}(f_{\theta}(\boldsymbol{x}))$ represents the entropy of the predicted distribution.

It can be seen from Eq. 1 and Eq. 2 that the objective of regularization-based calibration algorithms involves minimizing the classification loss while maximizing the predictive entropy. Specifically, the first objective typically represents a common classification loss (e.g., cross-entropy loss) that minimizes the discrepancy between $f_{\boldsymbol{\theta}}(\boldsymbol{x})$ and $\boldsymbol{y}$. The second objective serves as the regularization term, which prevents overconfidence of the deep neural network by encouraging lower confidence. For instance, in Eq. 1, the second term aims to push the predicted distribution closer to the uniform distribution, while in Eq. 2, it aims to maximize the entropy of the predicted distribution.

However, the above objectives faces a conflict, i.e., minimizing the classification loss by *increasing the confidence* corresponding to the ground-truth label and maximizing the entropy of the predictive probability by *decreasing the confidence*. To increase and decrease confidence are two opposite goals and striking the appropriate balance between them is difficult, leading to potential problems. For example, the model obtained with weak-regularization may classify all samples with high confidence, leading to overconfident especially on the challenging samples. Besides, strong regularization try to reduce confidence even on the easy samples, which may lead to an underconfident model especially on the easy samples, even harming the overall model performance (Wang et al., 2021). The key reason for these problems with previous methods is that they apply the same regularization to all samples without considering the inherent difficulty in correctly classifying each training sample.

## 3.2 Dynamic Regularization Calibration

In this section, we propose to leverage the inherent outliers in the training set to provide clearer guidance on how the model should adjust its predictive confidence on each sample. Intuitively, we recognize that strong regularization is not required for easy samples, which should be classified correctly with higher confidence. Conversely, for outlier samples in the training set, which are challenging to correctly classify, a lower confidence score should be assigned. This motivates us to calibrate confidence while accounting for the existence of outliers.

To this end, we assume the training set drawn from distribution $P$ follows Huber's $\eta$-contamination model (Huber, 1992), i.e.,

$$P = (1 - \eta)P_{in} + \eta P_{out}, \tag{3}$$

where $P_{in}$ and $P_{out}$ represent the in-distribution and arbitrary outlier distribution respectively. $0 < \eta < \frac{1}{2}$ represents the fraction of the outlier data. In practical scenarios, it is common that training data contains a mixture of simple in-distribution samples and challenging outlier samples. These simple in-distribution samples are more likely to be classified correctly, while some difficult

samples may not be classified correctly owing to the lack of crucial features or inherent ambiguity and indistinguishability (Seedat et al., 2022; Lorena et al., 2019). Empirical studies have also shown that even widely used datasets like ImageNet, CIFAR10, and CIFAR100 contain many easily classifiable samples as well as outlier samples that pose challenges for classification (Vasudevan et al., 2022; Toneva et al., 2018). Besides, the data augmentation methods, a widely-used component in the training of deep neural network, create novel samples by perturbing the training data, potentially introducing outlier samples (Yun et al., 2019; Cubuk et al., 2020). Considering the existence of outlier samples in the training data, we expect the model to be robust calibrated not only on in-distribution $P_{in}$, but also on outlier distribution $P_{out}$.

Unlike previous methods that regularize all samples indiscriminately, we impose dynamic regularization on the samples by considering the existence of outlier samples to provide explicit confidence supervision. Specifically, we first introduce the probability that a training sample $\boldsymbol{x}$ belongs to the in-distribution $P_{\boldsymbol{x} \sim P_{in}}$ or outlier distribution $P_{\boldsymbol{x} \sim P_{out}}$, where $P_{\boldsymbol{x} \sim P_{out}} + P_{\boldsymbol{x} \sim P_{in}} = 1$. Then we formally define the following dynamic regurlarization calibration loss $\mathcal{L}_{\text{DRC}}$:

$$\mathcal{L}_{\text{DRC}}(f_{\boldsymbol{\theta}}(\boldsymbol{x}), \boldsymbol{y}) = P_{\boldsymbol{x} \sim P_{in}} \mathcal{L}_{in}(f_{\boldsymbol{\theta}}(\boldsymbol{x}), \boldsymbol{y}) + P_{\boldsymbol{x} \sim P_{out}} \mathcal{L}_{out}(f_{\boldsymbol{\theta}}(\boldsymbol{x}), \boldsymbol{y}), \tag{4}$$

where $\mathcal{L}_{in}$ represents the in-distribution data loss, and in practice we can directly utilize the cross-entropy loss or square loss. $\mathcal{L}_{out}$ is the loss function corresponding to outliers, serving as a regularizer to make the model having lower confidence on outlier samples. Eq. 4 has an intuitive motivation that model should admit some samples are within its ability and should be classified accurately, while others are outside its ability. Specifically, for samples with a high probability of being in-distribution data, we should minimize its classification loss to increase the confidence. Conversely, for samples with a high probability of belonging to the outlier distribution, we should regularize its prediction confidence to avoid making overconfident decisions. By applying the dynamic regularization, we can achieve confident predictions on in-distribution samples, while avoiding overconfidence on challenging outlier samples.

However directly estimating the probability $P_{\boldsymbol{x} \sim P_{in}}$ and $P_{\boldsymbol{x} \sim P_{out}}$ presented in Eq. 4 is intractable since which distribution the training example is from is unknown. Therefore, we first present a simplified implementation of Eq. 4 and then show that this simplified version implicitly estimates $P_{\boldsymbol{x} \sim P_{in}}$ and $P_{\boldsymbol{x} \sim P_{out}}$ to achieve the same effect as the original objective when consider the whole training process. Specifically, in practice, given $B$ training examples during each training step, the simplified dynamic regurlarization calibration loss can be formulated as follows:

$$\widetilde{\mathcal{L}}_{\text{DRC}}(f_{\boldsymbol{\theta}}(\boldsymbol{x}_i), \boldsymbol{y}_i) = \delta_i \mathcal{L}_{ce}(f_{\boldsymbol{\theta}}(\boldsymbol{x}_i), \boldsymbol{y}_i) + (1 - \delta_i)\beta \mathcal{L}_{kl}(f_{\boldsymbol{\theta}}(\boldsymbol{x}_i), \boldsymbol{p}_u). \tag{5}$$

$\delta_i$ can be seen as whether the sample is in-distribution data and it is set to 0 if cross-entropy loss of $\boldsymbol{x}_i$ ranks in the top-$\eta B$ in a batch of $B$ samples, otherwise it is set to 1 , where $\eta$ is outlier fraction hyperparameter when the outlier proportion is unknown. $\mathcal{L}_{ce}(f_{\boldsymbol{\theta}}(\boldsymbol{x}_i), \boldsymbol{y}_i)$ is the cross-entropy loss serving as loss function of potential in-distribution data. $\mathcal{L}_{kl}(f_{\boldsymbol{\theta}}(\boldsymbol{x}_i), \boldsymbol{p}_u)$ denotes the KL-divergence between $f_{\boldsymbol{\theta}}(\boldsymbol{x}_i)$ and $\boldsymbol{p}_u$, serving as a regularizer to prevent overfitting on the potential outlier samples. $\beta$ is a hyperparameter to tune the strength of the KL-divergence. Eq. 5 simplifies the concept presented in Eq. 4 by performing non-parametric binary classification to avoid directly estimating the intractable $P_{\boldsymbol{x} \sim P_{in}}$ and $P_{\boldsymbol{x} \sim P_{out}}$. But when considering the whole training process, $P_{\boldsymbol{x} \sim P_{in}}$ and $P_{\boldsymbol{x} \sim P_{out}}$ can be implicitly estimated and Eq. 5 can have the similar objective to Eq. 4. Specifically, if sample $\boldsymbol{x}_i$ is sampled $S$ times during the whole training process, and at the $s$-th sampling $\boldsymbol{x}_i$ is categorized as an outlier ($\delta_i^s = 0$) or in-distribution data ($\delta_i^s = 1$). Then after $S$ times samplings, the whole objective for $\boldsymbol{x}_i$ can be written as

$$\sum_{s=1}^{S}[\delta_i^s \mathcal{L}_{ce}(f_{\boldsymbol{\theta}}(\boldsymbol{x}_i), \boldsymbol{y}_i) + (1 - \delta_i^s)\beta \mathcal{L}_{kl}(f_{\boldsymbol{\theta}}(\boldsymbol{x}_i), \boldsymbol{y}_i)]. \tag{6}$$

Then $P_{\boldsymbol{x} \sim P_{in}}$ and $P_{\boldsymbol{x} \sim P_{out}}$ are implicitly estimated as $\sum_{s=1}^{S} \delta_i^s / S$ and $1 - \sum_{s=1}^{S} \delta_i^s / S$ respectively. The whole pseudocode of the DRC is shown in Alg. 1.

## 4 EXPERIMENTS

We conduct extensive experiments on multiple datasets with outliers to answer the following questions. Q1 Effectiveness: Does DRC outperform other methods in terms of accuracy? Q2 Reliability: Can DRC obtain a more reliable model? Q3 Robustness: How does DRC perform on outlier data? Q4 Ablation study: How would the performance be if outlier samples are not exploited? Q5 Hyperparameter analysis: How do the hyperparameters in DRC affect model performance?

---

**Algorithm 1:** The training pseudocode of DRC.

---

**Input:** Training dataset $\mathcal{D}$, outlier fraction $\eta$ and hyperparameter $\beta$;
**Output:** The trained neural network $f_{\boldsymbol{\theta}}$.
**for** *each iteration* **do**

    Sample $B$ training samples $\{\boldsymbol{x}_i, \boldsymbol{y}_i\}_{i \in \mathbb{B}}$ from the training set $\mathcal{D}$ with a random index set $\mathbb{B}$;

    Compute the corresponding cross-entropy loss $\mathcal{L}_{ce}(f_{\boldsymbol{\theta}}(\boldsymbol{x}_i), \boldsymbol{y}_i)$ of each sample;

    Sort the losses and set $\delta_i$ for each sample according to the sorting result;

    Compute the loss $\widetilde{\mathcal{L}}_{\text{DRC}}(f_{\boldsymbol{\theta}}(\boldsymbol{x}_i), \boldsymbol{y}_i)$ according to Eq. 5 for each sample;

    Update $\boldsymbol{\theta}$ by one step to minimize $\mathbb{E}_{i \in \mathbb{B}}[\widetilde{\mathcal{L}}_{\text{DRC}}(f_{\boldsymbol{\theta}}(\boldsymbol{x}_i), \boldsymbol{y}_i)]$ with some gradient method.

---

## 4.1 EXPERIMENTAL SETUP

We briefly describe the experimental setup including the used experimental datasets, evaluation metrics, experimental settings and comparison methods. Details can be found in the appendix (Sec. B).

**Datasets.** We conduct extensive experiments on multiple datasets with potential outliers, including CIFAR-8-2, CIFAR-80-20 (Krizhevsky et al., 2009), Food101 (Bossard et al., 2014), Camelyon17 (Bandi et al., 2018; Koh et al., 2021), and ImageNetBG (Xiao et al., 2020). To evaluate model performance when datasets contain a certain fraction of outlier samples, the CIFAR-8-2 and CIFAR-80-20 datasets are constructed from CIFAR10 and CIFAR100 respectively. Specifically, we randomly select 8 classes from CIFAR10 (80 for CIFAR100) as in-distribution samples, while the remaining samples are randomly relabeled as one of the selected classes to serve as outliers. Food101 is a widely used classification dataset. Due to the imperfect data collection, the training set contains outlier samples, while the samples in the test set are all manually reviewed to ensure that they are all in-distribution data. Camelyon17 is a pathological image classification dataset consisting of multiple subgroups, which may contain challenging outlier data that cannot be accurately classified. ImageNetBG is a subset of ImageNet, which is used to evaluate the dependence of classifiers on the background of images. Its test set has both in-distribution data and challenging outliers with different predefined changes in the background of images.

**Evaluation metrics.** Following the standard metrics used in previous works (Moon et al., 2020; Corbière et al., 2019), we evaluate the performance from three perspectives: (1) the accuracy (ACC) on the test set; (2) ordinal ranking based confidence evaluation, including area under the risk coverage curve (AURC), excess-AURC (EAURC) (Geifman et al., 2018), false positive rate when the true positive rate is 95% (FPR95%TPR), and area under precision-recall curve with incorrectly classified examples as the positive class (AUPRErr) (Corbière et al., 2019); (3) calibration-based confidence evaluation, including expected calibration error (ECE) (Guo et al., 2017), the Brier score (Brier) (Brier, 1950) and negative log likelihood (NLL). For datasets where the test set consists of a mixture of ID and outlier samples, we show both the performance on all test samples and the performance on only outlier samples. For the CIFAR-8-2 and CIFAR-80-20 datasets, since the labels of the outlier samples are set randomly, we only report the calibration-based confidence evaluation metrics.

**Experimental settings.** We perform experiments under two different settings of weak augmentation and strong augmentation to validate the effectiveness of DRC. In this way, we can fully explore how outlier samples introduced by different augmentation techniques affect the calibration methods. Specifically, under the weak augmentation setting, same as standard setup of deep neural network training (He et al., 2016; Zagoruyko & Komodakis, 2016), we employ basic augmentation methods such as random cropping and image flipping. Under the strong augmentation setting, we use more aggressive methods, including random augmentation (Cubuk et al., 2020) and cutout (DeVries & Taylor, 2017). For all methods, under the exactly same setting, we tune hyperparameters based on the accuracy of validation set and run over three times to report the means and standard deviations.

**Comparison methods.** We conduct comparative experiments with multiple baseline methods, including empirical risk minimization (ERM), penalizing confidence (PC) (Pereyra et al., 2017), label smoothing (LS) (Müller et al., 2019), focal loss (FL) (Mukhoti et al., 2020), sample dependent focal loss (FLSD) (Mukhoti et al., 2020), inverse focal loss (IFL) (Wang et al., 2021), dual focal loss (Tao et al., 2023). Besides we conduct ablation study by removing outliers with high probability (RO) during training. Details of the comparison methods are in the appendix (Sec. B.4).

Table 1: The main experimental results (reported as percentages (%)) under the weak augmentation setting. The best and second-best results are in **bold** and underline, respectively. For datasets containing both ID and outlier test samples, we report the results on the full test set and the outlier subset. We mark whether the test samples are from the full set, ID subset, or outlier subset (Out.).

| Method | CIFAR-8-2 (Full) | | | (Out.) | ImageNetBG (Full) | | | Food101 (ID) | | |
|---|---|---|---|---|---|---|---|---|---|---|
| | ACC | EAURC | ECE | ECE | ACC | EAURC | ECE | ACC | EAURC | ECE |
| ERM | 77.86 | 1.84 | 14.17 | 55.69 | 85.79 | 1.23 | 4.79 | 84.99 | 1.62 | 4.82 |
| PC | 77.36 | 2.67 | 18.06 | 70.60 | 85.57 | 1.33 | 8.62 | 85.29 | 1.63 | 8.12 |
| LS | 77.93 | 5.01 | 8.37 | 36.33 | **86.62** | 1.34 | 10.01 | 85.04 | 2.21 | 10.49 |
| FLSD | 76.79 | 2.96 | 11.63 | 57.10 | 85.36 | 1.43 | 6.30 | 85.00 | 1.75 | 3.54 |
| FL | 77.26 | 2.37 | 16.31 | 67.07 | 85.67 | 1.29 | 1.50 | 85.37 | 1.68 | 1.32 |
| IFL | 78.30 | 1.02 | 5.64 | 14.01 | 85.47 | 1.26 | 6.67 | 86.50 | 1.52 | 7.64 |
| DFL | 77.14 | 2.35 | 16.00 | 65.82 | 85.66 | 1.42 | 1.62 | 85.50 | 1.63 | **0.80** |
| RO | 77.57 | 6.83 | 18.52 | 74.56 | 85.95 | 2.73 | 8.88 | 85.42 | 2.72 | 6.24 |
| DRC | **78.39** | **0.90** | **3.82** | **10.24** | 86.12 | **0.89** | **1.04** | **86.53** | **1.46** | 3.66 |

| Method | CIFAR-80-20 (Full) | | | (Out.) | ImageNetBG (Out.) | | | Camelyon17 (Out.) | | |
|---|---|---|---|---|---|---|---|---|---|---|
| | ACC | EAURC | ECE | ECE | ACC | EAURC | ECE | ACC | EAURC | ECE |
| ERM | 60.77 | 4.60 | 23.79 | 64.26 | 81.51 | 2.09 | 6.32 | 85.75 | 3.41 | 8.73 |
| PC | 62.10 | 4.44 | 23.77 | 64.50 | 81.24 | 2.23 | 11.20 | 85.16 | 3.42 | 11.44 |
| LS | 62.08 | 3.90 | **8.31** | 29.65 | **82.60** | 2.17 | 9.87 | 84.73 | 4.41 | 13.21 |
| FLSD | 59.44 | 5.04 | 12.92 | 45.59 | 80.94 | 2.41 | 6.32 | 86.09 | 3.69 | 9.12 |
| FL | 60.30 | 4.96 | 18.30 | 53.16 | 81.36 | 2.18 | 2.08 | 86.60 | 3.36 | 3.45 |
| IFL | 61.82 | 3.62 | 20.80 | 46.29 | 81.11 | 2.14 | 8.69 | 85.85 | 2.92 | 11.21 |
| DFL | 59.94 | 4.96 | 17.24 | 52.10 | 81.34 | 2.38 | 1.75 | 85.75 | 3.03 | **2.87** |
| RO | 60.78 | 13.92 | 24.25 | 65.14 | 81.82 | 4.49 | 11.52 | 84.36 | 5.83 | 12.34 |
| DRC | **62.74** | **3.19** | 9.53 | **15.38** | 81.97 | **1.51** | **1.32** | **87.46** | **2.83** | 6.39 |

## 4.2 EXPERIMENTAL RESULTS

We conduct experiments to answer the above-posed questions. The main experimental results under different settings are presented in Tab. 1 and Tab. 2. Detailed experimental results with standard diversion and more evaluation metrics are shown in the Tab. 3 and Tab. 4 of the appendix.

**Q1 Effectiveness.** Compared to other methods, DRC achieves superior performance in terms of accuracy. Specifically, as shown in the experimental results, compared with previous methods, DRC achieves top two accuracy rankings consistently on almost all datasets under the weak-augmentation and strong augmentation settings. For example, on the Camelyon17 dataset, DRC achieves the best accuracy performance of 87.46% and 93.43% under weak and strong augmentation settings. This improvement can be attributed to the ability of DRC to prevent the neural network from overfitting to outlier samples in the training data, thereby improving the generalization and classification accuracy on the test data slightly. Although DRC demonstrates superior performance, please note that the main goal of calibration methods is to calibrate the model thereby improving trustworthiness rather than improve accuracy simply.

**Q2 Reliability.** We can obtain the following observations from the experimental results. (1) DRC can obtain the state-of-the-art confidence quality in terms of ranking-based metrics (e.g., EAURC) across almost all datasets. Specifically, under the weak-augmentation setting, DRC achieves superior EAURC performance on all datasets. For example, on the CIFAR-80-20 dataset, DRC outperforms the second best method by 0.43% in terms of EAURC. Moreover, under the strong-augmentation setting, DRC achieves the best performance across all datasets except Food101 dataset. (2) DRC also demonstrates outstanding performance on calibration-based metrics (e.g., ECE). For example, under weak-augmentation setting, DRC achieves 3.82% performance on the full test set of CIFAR-8-2 dataset, which is 1.82% lower than the second best method. The key reason behind the performance improvements is that DRC effectively leverages outlier samples in the training set to provide more explicit confidence supervision to improve the confidence quality, resulting in more reliable predictions.

Table 2: The main experimental results (reported as percentages (%)) under the strong augmentation setting. The best and second-best results are in **bold** and underline, respectively. For datasets containing both ID and outlier test samples, we report the results on the full test set and the outlier subset. We mark whether the test samples are from the full set, ID subset, or outlier subset (Out.).

| Method | CIFAR-8-2 (Full) | | | (Out.) | ImageNetBG (Full) | | | Food101 (ID) | | |
|---|---|---|---|---|---|---|---|---|---|---|
| | ACC | EAURC | ECE | ECE | ACC | EAURC | ECE | ACC | EAURC | ECE |
| ERM | 79.35 | 0.70 | 6.89 | 26.42 | 87.35 | 0.98 | 3.47 | 87.26 | 1.25 | 2.37 |
| PC | 79.65 | 0.64 | 4.94 | 15.17 | 86.88 | 1.09 | 7.19 | 87.54 | 1.24 | 5.87 |
| LS | 79.65 | 4.57 | 10.01 | 39.34 | **87.54** | 1.26 | 4.42 | 87.33 | 1.80 | 19.70 |
| FLSD | 78.22 | 2.33 | 8.58 | 52.43 | 86.64 | 1.21 | 7.51 | 85.98 | 1.55 | 6.04 |
| FL | 78.96 | 1.07 | 9.51 | 43.23 | 86.70 | 1.17 | 3.92 | 86.32 | 1.45 | 3.62 |
| IFL | 79.70 | 0.70 | 4.97 | 14.40 | 87.26 | 1.03 | 4.89 | **87.67** | **1.22** | 5.05 |
| DFL | 78.69 | 1.97 | 13.25 | 60.51 | 87.11 | 1.17 | 2.96 | 86.80 | 1.40 | **1.29** |
| RO | 75.38 | 5.48 | 17.36 | 58.97 | 87.21 | 2.08 | 6.22 | 87.23 | 2.12 | 3.04 |
| DRC | **79.97** | **0.62** | **3.30** | **8.81** | 87.53 | **0.94** | **2.23** | 87.32 | 1.25 | 2.17 |

| Method | CIFAR-80-20 (Full) | | | (Out.) | ImageNetBG (Out.) | | | Camelyon17 (Out.) | | |
|---|---|---|---|---|---|---|---|---|---|---|
| | ACC | EAURC | ECE | ECE | ACC | EAURC | ECE | ACC | EAURC | ECE |
| ERM | 63.68 | 3.42 | 17.08 | 47.87 | 83.52 | 1.68 | 4.61 | 90.21 | 1.41 | 4.60 |
| PC | 64.10 | 3.02 | 16.09 | 37.96 | 82.91 | 1.83 | 9.40 | 87.30 | 1.97 | 9.91 |
| LS | 63.73 | 4.17 | **5.07** | 30.26 | 83.81 | 2.07 | 4.21 | 92.52 | 1.28 | 18.69 |
| FLSD | 61.88 | 5.02 | 16.49 | 55.18 | 82.63 | 2.05 | 7.92 | 92.24 | 1.12 | 12.81 |
| FL | 63.06 | 3.30 | 9.29 | 30.87 | 82.72 | 1.99 | 4.04 | 92.30 | 1.11 | 12.91 |
| IFL | 64.43 | 3.02 | 21.66 | 54.86 | 83.41 | 1.73 | 6.43 | 88.99 | 1.59 | 7.15 |
| DFL | 63.47 | 3.31 | 8.42 | 29.68 | 83.28 | 1.97 | 3.03 | 90.60 | 1.72 | 7.18 |
| RO | 60.20 | 14.23 | 24.73 | 65.26 | 83.41 | 3.51 | 8.14 | 91.55 | 2.99 | 7.10 |
| DRC | **65.90** | **2.37** | 9.84 | **20.57** | **83.83** | **1.58** | **2.97** | **93.43** | **0.83** | **2.30** |

**Q3 Robustness.** To evaluate the robustness of DRC, we also show the performance of the different method on the outlier test dataset. We can draw the following observations. (1) When evaluated on outlier datasets, the performance usually decreases. For example, on the outlier subset of the ImageNetBG test set, the performance is lower across all metrics compared to the full test set. This highlights the necessity to study robust calibration methods. (2) DRC shows excellent performance on outlier test dataset. For example, under the weak augmentation setting, DRC achieves 10.24% and 15.38% in terms of ECE on the outlier subset of CIFAR-8-2 and CIFAR-80-20, outperforming the second best by 3.77% and 14.27% respectively. This indicates that DRC is more robust to potential outlier data, because it effectively utilizes the outlier data during training to impose clear guidance about what should be unknown for the model.

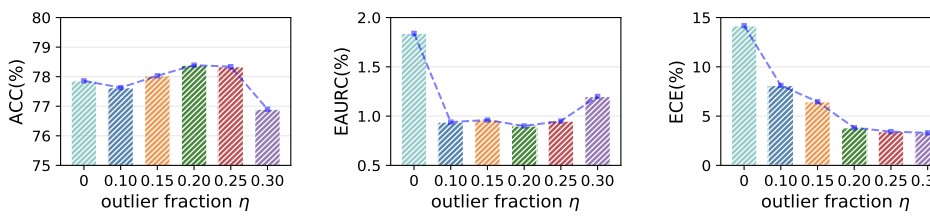

Figure 2: Performance of the model on different metrics with varying outlier fraction hyperparameter $\eta$, while fixing the regularization strength $\beta$ to 1, on the CIFAR-8-2 dataset.

**Q4 Ablation study.** As shown in Tab. 1 and Tab. 2, compared with ERM and removing outliers (RO), DRC consistently shows better performance. Specifically, our experiments demonstrate that utilizing outlier samples for regularization during deep neural network training through DRC can lead to more reliable confidence estimation than ERM and RO.

**Q5 Hyperparameter Analysis.** We conduct hyperparameter analysis on the CIFAR-8-2 dataset under the weak augmentation setting. To evaluate the effect of the outlier fraction hyperparameter $\eta$, we tune $\eta$ and show the corresponding experimental results in Fig. 2. From the experimental results, we can have the following observations: (1) when the outlier fraction hyperparameter $\eta$ is close to the true outlier fraction (0.2 for CIFAR-8-2), DRC can achieve better performance in terms of accuracy and EAURC; (2) increasing the outlier fraction hyperparameter $\eta$ can obtain better model in terms of ECE. However, when it exceeds the true outlier fraction, the improvement of ECE performance is limited. More hyperparameter analysis results are provided in the appendix (Sec. B.6).

## 5 THEORETICAL ANALYSIS

In this section, we aim to characterize the calibration error of DRC and the previous regularization-based methods under the Huber's $\eta$-contamination model (Eq. 3). The results demonstrate that DRC could obtain smaller calibration error.

**Data generative model.** We consider the Huber's $\eta$-contamination model described in Eq. 3. Specifically, we assume $P_{in}$ follows a Gaussian mixture model for binary classification, where $X$ is standard Gaussian, $Y \in \{-1, 1\}$ with prior probability $\mathbb{P}(Y = 1) = \mathbb{P}(Y = -1)$, and

$$X \mid Y \sim N(Y \cdot \boldsymbol{w}^*, I_d),$$

where $I_d$ denotes the $d$-dimensional identity matrix, $\boldsymbol{w}^*$ is the ground-truth coefficient vector. We further assume $P_{out}$ follows the opposite distribution where $X \mid Y \sim N(-Y \cdot \boldsymbol{w}^*, I_d)$. We have i.i.d. observations $\{(\boldsymbol{x}_i, y_i)\}_{i=1}^n$ sampled from the distribution $P = (1 - \eta)P_{in} + \eta P_{out}$.

**The baseline estimator.** We consider the method of minimizing the common used classification loss (e.g., square loss) with label smoothing as our baseline, which produces a solution:

$$\hat{\boldsymbol{w}} = \arg\min_{\boldsymbol{w}} \frac{1}{n} \sum_{i=1}^n (\boldsymbol{w}^\top \boldsymbol{x}_i - \tilde{y}_i)^2,$$

where $\tilde{y}_i = (1 - \epsilon)y_i + \epsilon/2$. For $k \in \{-1, 1\}$, the confidence $\hat{\mathbb{P}}_k(\boldsymbol{x})$ is an estimator of $\hat{\mathbb{P}}(y = k|\boldsymbol{x})$, and it takes the form $\hat{\mathbb{P}}_k(\boldsymbol{x}) = 1/(e^{-k \cdot \hat{\boldsymbol{w}}^\top \boldsymbol{x}} + 1)$. Other regularization-based methods essentially have similar goals to label smoothing.

**Calibration error.** Here we consider the case where $\mathbb{P}_1(X) > 1/2$, as the case where $\mathbb{P}_1(X) \leq 1/2$ can be analyzed similarly by symmetry. For $p \in (1/2, 1)$, the signed calibration error at a confidence level $p$ is $p - \mathbb{P}(Y = 1 \mid \hat{\mathbb{P}}_1(X) = p)$. As a result, the formula of calibration error (ECE) is given by

$$ECE[\hat{\mathbb{P}}] = \mathbb{E}[|\mathbb{P}(Y|\hat{\mathbb{P}}(X) = p) - p|].$$

In the following, we show that the calibration error of our proposed Algorithm 1, denoted by $\hat{\mathbb{P}}_{\text{DRC}}$, is smaller than the baseline algorithm $\hat{\mathbb{P}}_{baseline}$. The proof is in the appendix (Sec. 1).

**Theorem 1.** *Consider the data generative model and the learning setting above. We assume $\|\boldsymbol{w}^*\| \leq c_0$ for some sufficiently small $c_0 > 0$, and $d/n = o(1)$. Suppose the initialization parameter $\boldsymbol{\theta}^{(0)}$ satisfies $\|\boldsymbol{\theta}^{(0)} - \boldsymbol{w}^*\| \leq c_1$ for a sufficiently small constant $c_1 > 0$. Then, for sufficiently large $n$, for $k = 2, \ldots, K$, we have*

$$ECE[\hat{\mathbb{P}}_{\text{DRC}}] < ECE[\hat{\mathbb{P}}_{baseline}].$$

## 6 CONCLUSION

In this paper, we summarize the core principle of existing regularization-based calibration methods, and show their underlying limitations due to lack of explicit confidence supervision. To address these limitations, we propose a simple yet effective approach called dynamic regularization calibration (DRC), which regularizes the model using potential outlier samples in the training data, thus allowing us to provide the model with clear direction on confidence calibration by informing the model what it should know and what it should not know. DRC significantly outperforms existing methods on real-world datasets, achieving robust calibration performance. Moreover, the theoretical analyses shows that DRC achieves smaller calibration error over previous methods. In this work, we first introduce the paradigm of dynamic regularization for calibration and provide a simple yet effective implementation. In the future, we believe exploring more elegant and effective strategies for dynamic regularization will be an interesting and promising direction.

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

# Appendix

## A    REGULARIZATION-BASED CALIBRATION METHODS

**Evidential deep learning** (Sensoy et al., 2018) constructs the classification outputs as Dirichlet distributions $Dir(\boldsymbol{\alpha})$ with parameters $\boldsymbol{\alpha} = [\alpha^1, \cdots, \alpha^K]$, and then minimizes the expected distance between the obtained Dirichlet distributions and the labels while regularizing by minimizing the KL divergence between the obtained Dirichlet distributions and the uniform distribution. The loss function is formulated as follows:

$$\sum_{k=1}^{K} y^k \left( \psi(S) - \psi(\alpha^k) \right) + \gamma KL[Dir(\tilde{\boldsymbol{\alpha}}), Dir([1, \cdots, 1])], \tag{7}$$

where $S = \sum_{k=1}^{K} \alpha^k$ is the Dirichlet strengthes, $\psi(\cdot)$ represents the digamma function, $\gamma$ is a hyperparameter, $\tilde{\boldsymbol{\alpha}} = \boldsymbol{y} + (1 - \boldsymbol{y}) \odot \boldsymbol{\alpha}$, and $Dir([1, \cdots, 1])$ is a uniform Dirichlet distribution.

**Penalizing confidence** (Pereyra et al., 2017) suggest a confidence penalty term to prevent the deep neural networks from overfitting and producing overconfident predictions. Formally, the loss function of penalizing confidence is defined as:

$$\mathcal{L}_{ce}(f_\theta(\boldsymbol{x}), \boldsymbol{y}) - \gamma \mathcal{H}(f_\theta(\boldsymbol{x})), \tag{8}$$

where $\gamma$ is a hyperparameter to control the penalizing strength.

## B    EXPERIMENTAL DETAILS

In this section, we present the experimental setup in detail including the backbone model for each dataset (Sec. B.1), descriptions of the datasets (Sec. B.2), evaluation metrics (Sec. B.3), comparison methods (Sec. B.4), comparison experimental results on various metrics (Sec. B.5), and additional experimental results from hyperparameter analysis (Sec. B.6). We are committed to open-sourcing the code related to our research after publication to present more details.

### B.1    BACKBONE MODEL

For the CIFAR-8-2 and CIFAR-80-20 datasets, we use a randomly initialized WideResNet-28-10 (Zagoruyko & Komodakis, 2016) as the backbone network; for the Camelyon17 dataset, we use a DenseNet-121 (Huang et al., 2017) network pre-trained on ImageNet as the backbone network; for other datasets, we use a ResNet-50 (He et al., 2016) network pre-trained on ImageNet as the backbone network.

### B.2    DATASETS DETAILS

The datasets used in the experiments are described in detail here.

- `CIFAR-8-2`: The CIFAR-8-2 dataset is artificially constructed to evaluate the performance of the model when the outlier fraction of the dataset is available. Specifically, we randomly select 8 classes from CIFAR10 (Krizhevsky et al., 2009) as in-distribution samples, while the remaining samples are randomly relabeled as one of the selected classes to serve as outliers. The true outlier fraction of the CIFAR-8-2 dataset is 20%. Since the outlier labels in the cifar-8-2 dataset are randomly generated, the accuracy and ordinal ranking based confidence evaluation metrics lose their meaning on this dataset. Therefore, we do not report these metrics for the outlier dataset in the experimental results.

- `CIFAR-80-20`: Samilar as CIFAR-8-2, we randomly select 80 classes from CIFAR100 (Krizhevsky et al., 2009) dataset, and relabel other samples as one of the selected calsses to serve as outliers. The true outlier fraction of the CIFAR-80-20 dataset is 20%. Since the outlier labels in the cifar-80-20 dataset are randomly generated, the accuracy and ordinal ranking based confidence evaluation metrics lose their meaning on this dataset. Therefore, we do not report these metrics for the outlier dataset in the experimental results.

- `Camelyon17`: Camelyon17 is a pathology image dataset containing over 450,000 lymph node scans from 5 different hospitals, used for detecting cancerous tissues in images (Bandi et al., 2018). Similar to previous work (Koh et al., 2021), we take part of the data from 3 hospitals as the training set. The remaining data from these 3 hospitals, together with data from another hospital, are used as the validation set. The last hospital is used as an outlier test set. Notably, due to differences in pathology staining methods between hospitals, even data within the same hospital can be seen as sampled from multiple subpopulations. We verify on the Camelyon17 dataset whether models can achieve more robust generalization performance when the training set contains multiple subgroups.

- `ImageNetBG`: ImageNetBG is a benchmark dataset for evaluating the dependence of classifiers on image backgrounds (Xiao et al., 2020). It consists of a 9-class subset of ImageNet (ImageNet-9) and provides bounding boxes that allow removing the background. Similar as in previous settings (Yang et al., 2023), we train models on the original IN-9L (with background) set, adjust hyperparameters based on validation accuracy, and evaluate on the test set (in-distribution data), MIXED-RAND, NO-FG and ONLY-FG test set (outlier data).

- `Food101`: Food101 is a commonly used food classification dataset containing 101 food categories with a total of 101,000 images (Bossard et al., 2014). For each category, there are 750 training images and 250 manually verified test images. The training images are intentionally unclean and contain some amount of noise, primarily in the form of intense colors and occasionally wrong labels, which can be seen as outlier data.

## B.3 Evaluation metrics

The evaluation metrics used in the experiments are described in detail here.

- `AURC` and `EAURC`: The AURC is defined as the area under the risk-coverage curve (Geifman & El-Yaniv, 2017), where risk represents the error rate and coverage refers to the proportion of samples with confidence estimates exceeding a specified confidence threshold. A lower AURC indicates that correct and incorrect samples can be effectively separated based on the confidence of the samples. However, AURC is influenced by the predictive performance of the model. To allow for meaningful comparisons across models with different performance, Excess-AURC (E-AURC) is proposed by (Geifman et al., 2018) by subtracting the optimal AURC (the minimum possible value for a given model) from the empirical AURC.

- `AUPRErr`: AUPRErr represents the area under the precision-recall curve where misclassified samples (i.e., incorrect predictions) are used as positive examples. This metric can evaluate the capability of the error detector to distinguish between incorrect and correct samples. A higher AUPRErr usually indicates better error detection performance (Corbière et al., 2019).

- `FPR95%TPR`: The FPR95%TPR metric measures the false positive rate (FPR) when the true positive rate (TPR) is fixed at 95%. This metric can be interpreted as the probability that an incorrect prediction is mistakenly categorized as a correct prediction, when the TPR is set to 95%.

- `ECE`: The Expected Calibration Error (ECE) provides a measure of the alignment between the predicted confidence scores and labels. It partitions the confidence sores into multiple equally spaced intervals, computes the difference between accuracy and average confidence in each interval, and then aggregates the results weighted by the number of samples. Lower ECE usually indicates better calibration.

- `NLL`: The Negative Log Likelihood (NLL) measures the log loss between the predicted probabilities and the one-hot label encodings. Lower NLL corresponds to higher likelihood of the predictions fitting the true distribution.

- `Brier`: The Brier score calculates the mean squared error between the predicted probabilities and the one-hot label.

Since the outlier labels in the cifar-8-2 and cifar-80-20 datasets are randomly generated, the accuracy and ordinal ranking based confidence evaluation metrics lose their meaning on this datasets. Therefore, we do not report these metrics for the outlier dataset in the experimental results.

### B.4 COMPARISON METHODS

The Comparison methods are described in detail here.

- ERM trains the model by minimizing the empirical risk on the training data, using cross-entropy as the loss function.
- PC trains the model with cross-entropy loss while regularize the neural networks by penalizing low entropy predictions.
- LS is a regularization technique that trainin the neural network with softened target labels.
- FLSD refers to a sample-dependent focal loss, where the hyperparameters of the focal loss are set differently for samples with different confidence scores (Mukhoti et al., 2020).
- FL refers to focal loss, which implicitly regularize the deep neural network by increasing the weight of samples with lager losses.
- IFL conduct a simple modification on the weighting term of original focal loss by assigning larger weights to the samples with larger output confidences.
- DFL aims to achieve a better balance between over-confidence and under-confidence by maximizing the gap between the ground truth logit and the highest logit ranked after the ground truth logit.
- RO: We conduct ablation studies by removing potential outlier samples during training. Specifically, during training, given $B$ samples, we directly drop the top $\eta B$ samples with the highest losses, where $\eta$ is the predefined outlier fraction.

### B.5 ADDITIONAL RESULTS

In Tab. 3 and Tab. 4, we present the experimental results for all evaluation metrics along with the corresponding standard deviations. From the experimental results we can draw similar conclusions as those in the experiments section. Specifically, DRC achieves the best performance 87 times out of 108 experiments, which strongly suggest DRC outperforms previous approaches in calibration.

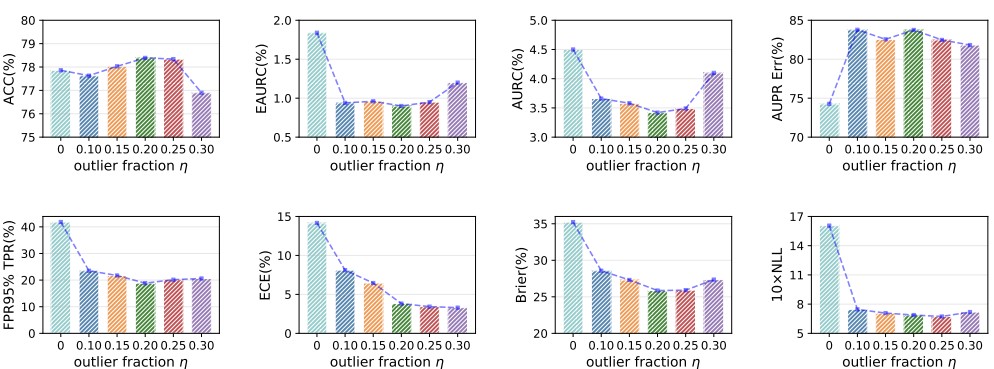

Figure 3: Performance of the model on multiple metrics with varying outlier fraction hyperparameter $\eta$, while fixing the regularization strength $\beta$ to 1, on the CIFAR-8-2 dataset.

### B.6 MORE RESULTS OF HYPERPARAMETER ANALYSIS

We present the detailed hyperparameter analysis results on the CIFAR-8-2 and CIFAR-80-20 datasets in Fig.3, Fig.4, Fig.5 and Fig.6. Specifically, to evaluate the effect of the outlier fraction hyperparameter $\eta$ and regularization strength $\beta$ on the model, we tune one hyperparameter while fixing the other. From the experimental results we can draw the following conclusions: (1) As shown in Fig.3 and Fig.5, when the set outlier fraction hyperparameter $\eta$ is close to the true outlier sample ratio, the model can achieve relatively optimal performance. Meanwhile increasing $\eta$ within a certain range does not significantly degrade the model performance. For example, on most metrics of the CIFAR-8-2 and CIFAR-80-20 datasets, the relatively best performance is achieved at $\eta = 0.2$. (2)

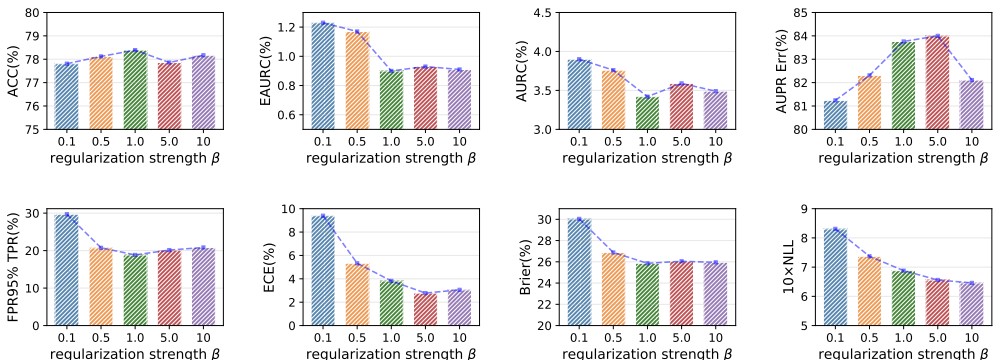

Figure 4: Performance of the model on different metrics with varying regularization strength hyperparameter $\beta$, while fixing the outlier fraction hyperparameter $\eta$ to 0.2, on the CIFAR-8-2 dataset.

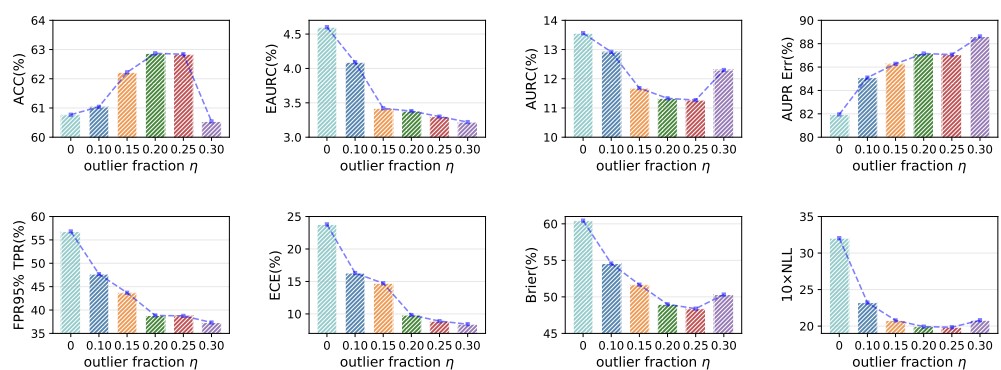

Figure 5: Performance of the model on multiple metrics with varying outlier fraction hyperparameter $\eta$, while fixing the regularization strength $\beta$ to 1, on the CIFAR-80-20 dataset.

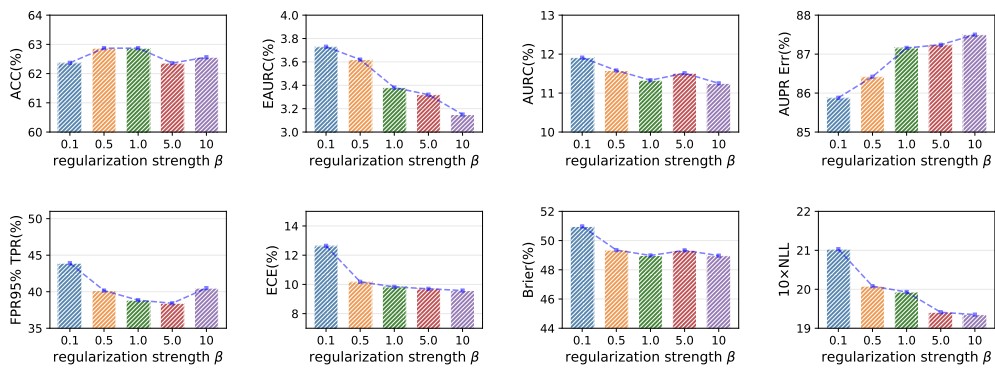

Figure 6: Performance of the model on different metrics with varying regularization strength hyperparameter $\beta$, while fixing the outlier fraction hyperparameter $\eta$ to 0.2, on the CIFAR-80-20 dataset.

As shown in Fig.4 and Fig.6, we can find that increasing the regularization strength $\beta$ to a certain level yields relatively good performance, after which further increases does not significantly improve the results. For instance, when $\beta$ exceeds 1, the performance of the model remains relatively stable, with most metrics changing negligibly.

Table 3: The comparison experimental results on different datasets and different methods under the weak augmentation setting. ↓ and ↑ indicate lower and higher values are better respectively. For better presentation, the best and second-best results are in **bold** and uderline respectively. For clarity, NLL values are multiplied by 10. Remaining values are reported as percentages (%). For datasets with both ID and outlier test samples, we report the results on all samples and outlier samples. We mark whether the test samples are sampled from ID or outliers in the table. *Compared with other methods,* DRC *achieves excellent performance on different metrics in almost all datasets.*

| Dataset | Method | ACC (↑) | EAURC (↓) | AURC (↓) | AUPR Err (↑) | FPR95% TPR(↓) | ECE (↓) | Brier (↓) | NLL (↓) |
|---|---|---|---|---|---|---|---|---|---|
| CIFAR -8-2 (All) | ERM | 77.86±0.19 | 1.84±0.66 | 4.50±0.71 | 74.27±5.70 | 41.81±14.15 | 14.17±5.27 | 35.22±5.55 | 16.06±6.34 |
| | PC | 77.36±0.14 | 2.67±1.08 | 5.45±1.09 | 71.12±6.43 | 48.10±12.39 | 18.06±2.45 | 39.34±3.32 | 23.63±9.21 |
| | LS | 77.93±0.23 | 5.01±2.95 | 7.65±2.99 | 74.82±6.05 | 34.36±9.55 | 8.37±1.02 | 30.08±2.61 | 7.66±0.65 |
| | FLSD | 76.79±0.12 | 2.96±0.16 | 5.89±0.18 | 68.88±1.13 | 56.54±1.52 | 11.63±0.07 | 35.42±0.26 | 11.87±0.15 |
| | FL | 77.26±0.25 | 2.37±0.05 | 5.18±0.12 | 70.58±0.18 | 52.30±0.88 | 16.31±0.13 | 37.65±0.30 | 15.38±0.21 |
| | IFL | 78.30±0.18 | 1.02±0.14 | 3.57±0.10 | 82.98±1.27 | **18.80±2.05** | 5.64±1.04 | 26.80±0.76 | 7.92±0.88 |
| | DFL | 77.14±0.16 | 2.35±0.07 | 5.19±0.05 | 69.92±1.06 | 53.53±1.44 | 16.00±0.09 | 37.65±0.24 | 15.48±0.07 |
| | RO | 77.57±0.31 | 4.10±0.44 | 6.83±0.40 | 65.33±0.89 | 57.23±0.75 | 18.52±0.47 | 39.97±0.65 | 18.55±0.60 |
| | DRC | **78.39±0.02** | **0.90±0.04** | **3.42±0.03** | **83.76±0.49** | 18.85±1.71 | **3.82±0.24** | **25.86±0.21** | **6.88±0.07** |
| CIFAR -80-20 (All) | ERM | 60.77±0.37 | 4.60±0.18 | 13.56±0.34 | 81.95±0.18 | 56.81±0.30 | 23.79±0.39 | 60.44±0.72 | 32.04±0.66 |
| | PC | 62.10±0.28 | 4.44±0.05 | 12.75±0.14 | 81.40±0.31 | 55.31±1.48 | 23.77±0.36 | 59.27±0.53 | 29.61±0.47 |
| | LS | 62.08±0.43 | 3.90±0.14 | 12.22±0.33 | 85.06±0.37 | 46.26±0.72 | **8.31±0.81** | 48.99±0.67 | **17.79±0.26** |
| | FLSD | 59.44±0.36 | 5.04±0.14 | 14.68±0.32 | 82.15±0.42 | 55.51±0.96 | 12.92±0.35 | 54.81±0.49 | 23.31±0.17 |
| | FL | 60.30±0.12 | 4.73±0.18 | 13.92±0.21 | 81.40±0.17 | 55.35±0.95 | 18.30±0.12 | 56.81±0.26 | 26.06±0.28 |
| | IFL | 61.82±0.46 | 3.62±0.03 | 12.07±0.25 | 86.66±0.31 | 44.96±0.63 | 20.80±0.49 | 56.18±0.62 | 27.76±0.25 |
| | DFL | 59.94±0.21 | 4.96±0.04 | 14.34±0.14 | 81.59±0.37 | 55.03±0.50 | 17.24±0.22 | 56.71±0.40 | 26.22±0.15 |
| | RO | 60.78±0.46 | 4.96±0.10 | 13.92±0.19 | 80.99±0.35 | 58.50±0.65 | 24.25±0.85 | 61.08±0.75 | 30.90±2.45 |
| | DRC | **62.74±0.44** | **3.19±0.07** | **11.21±0.24** | **87.41±0.21** | 40.02±0.92 | 9.53±0.41 | **48.76±0.71** | 19.31±0.13 |
| Image NetBG (All) | ERM | 85.79±0.11 | 1.23±0.01 | 2.29±0.01 | 62.29±0.37 | 47.97±0.50 | 4.79±0.41 | 20.30±0.20 | 4.71±0.08 |
| | PC | 85.57±0.15 | 1.33±0.07 | 2.43±0.08 | 62.42±1.70 | 49.01±3.92 | 8.62±0.46 | 22.45±0.71 | 6.30±0.44 |
| | LS | **86.62±0.12** | 1.34±0.03 | 2.27±0.05 | 62.91±0.28 | 43.95±0.46 | 10.01±0.30 | 19.84±0.09 | 4.92±0.03 |
| | FLSD | 85.36±0.16 | 1.43±0.05 | 2.56±0.03 | 61.70±1.63 | 49.68±2.06 | 6.30±0.08 | 21.00±0.04 | 4.81±0.01 |
| | FL | 85.67±0.11 | 1.29±0.02 | 2.37±0.03 | 62.62±0.19 | 48.03±0.25 | 1.50±0.11 | 19.76±0.12 | 4.42±0.03 |
| | IFL | 85.47±0.05 | 1.26±0.03 | 2.38±0.03 | 62.99±0.48 | 48.07±0.74 | 6.67±0.32 | 21.32±0.11 | 5.21±0.07 |
| | DFL | 85.66±0.25 | 1.42±0.07 | 2.51±0.11 | 61.69±0.49 | 48.87±2.11 | 1.62±0.07 | 19.94±0.38 | 4.50±0.09 |
| | RO | 85.95±0.18 | 1.70±0.03 | 2.73±0.01 | 58.32±0.63 | 52.66±0.28 | 8.88±0.18 | 22.56±0.29 | 6.44±0.08 |
| | DRC | 86.12±0.11 | **0.89±0.01** | **1.90±0.02** | **67.74±0.34** | 38.18±0.42 | **1.04±0.17** | **18.71±0.18** | **4.41±0.06** |
| Image NetBG (Outlier) | ERM | 81.51±0.15 | 2.09±0.02 | 3.91±0.02 | 62.91±0.36 | 56.87±0.50 | 6.32±0.52 | 26.34±0.30 | 6.12±0.11 |
| | PC | 81.24±0.17 | 2.23±0.12 | 4.11±0.14 | 63.03±1.70 | 57.12±2.79 | 11.20±0.57 | 29.14±0.93 | 8.18±0.58 |
| | LS | **82.60±0.16** | 2.17±0.04 | 3.78±0.07 | 63.60±0.31 | 51.89±0.65 | 9.87±0.39 | 25.24±0.10 | 6.07±0.03 |
| | FLSD | 80.94±0.20 | 2.41±0.07 | 4.36±0.06 | 62.41±1.60 | 57.78±1.66 | 6.32±0.13 | 26.88±0.06 | 6.06±0.02 |
| | FL | 81.36±0.13 | 2.18±0.03 | 4.04±0.05 | 63.19±0.16 | 56.30±0.58 | 2.08±0.25 | 25.61±0.15 | 5.71±0.05 |
| | IFL | 81.11±0.04 | 2.14±0.05 | 4.05±0.06 | 63.60±0.43 | 57.04±0.74 | 8.69±0.38 | 27.66±0.13 | 6.76±0.08 |
| | DFL | 81.34±0.35 | 2.38±0.12 | 4.24±0.19 | 62.33±0.46 | 56.63±1.86 | 1.75±0.36 | 25.79±0.53 | 5.79±0.12 |
| | RO | 81.82±0.24 | 2.73±0.03 | 4.49±0.03 | 59.33±0.70 | 59.97±0.51 | 11.52±0.22 | 29.13±0.38 | 8.33±0.10 |
| | DRC | 81.97±0.15 | **1.51±0.01** | **3.24±0.04** | **68.38±0.33** | 47.09±0.48 | **1.32±0.22** | **24.17±0.23** | **5.68±0.08** |
| Food 101 (ID) | ERM | 84.99±0.09 | 1.62±0.01 | 2.80±0.02 | 60.33±0.44 | 52.35±0.68 | 4.82±0.18 | 21.90±0.14 | 5.87±0.03 |
| | PC | 85.29±0.16 | 1.63±0.02 | 2.77±0.04 | 59.95±0.32 | 52.59±0.97 | 8.12±0.12 | 22.94±0.29 | 7.15±0.09 |
| | LS | 85.04±0.10 | 2.21±0.03 | 3.39±0.04 | 57.59±0.31 | 55.80±0.42 | 10.49±0.07 | 23.35±0.07 | 6.79±0.03 |
| | FLSD | 85.00±0.07 | 1.75±0.03 | 2.93±0.03 | 58.00±0.42 | 55.94±1.31 | 3.54±0.04 | 21.82±0.04 | 5.47±0.01 |
| | FL | 85.37±0.13 | 1.68±0.02 | 2.81±0.03 | 58.61±0.93 | 55.37±0.80 | 1.32±0.10 | 21.11±0.07 | 5.32±0.01 |
| | IFL | 86.50±0.12 | 1.52±0.03 | 2.47±0.05 | 57.81±0.20 | **51.32±0.37** | 7.64±0.16 | 21.29±0.26 | 6.66±0.12 |
| | DFL | 85.50±0.08 | 1.63±0.02 | 2.74±0.03 | 58.52±0.27 | 52.95±1.04 | **0.80±0.15** | 20.79±0.10 | **5.30±0.01** |
| | RO | 85.42±0.26 | 1.61±0.03 | 2.72±0.07 | 58.67±0.15 | 53.70±0.06 | 6.24±0.33 | 21.93±0.45 | 6.11±0.14 |
| | DRC | **86.53±0.09** | **1.46±0.02** | **2.41±0.01** | 57.93±0.17 | 51.61±0.89 | 3.66±0.14 | **19.95±0.05** | 5.44±0.02 |
| Came lyon (Outlier) | ERM | 85.75±1.32 | 3.41±0.66 | 4.48±0.86 | 39.42±0.39 | 73.35±2.54 | 8.73±1.15 | 22.56±2.32 | 4.41±0.54 |
| | PC | 85.16±0.47 | 3.42±0.36 | 4.58±0.43 | 40.42±0.61 | 74.48±1.54 | 11.44±0.43 | 25.41±0.93 | 6.38±0.32 |
| | LS | 84.73±1.43 | 4.41±0.71 | 5.65±0.94 | 38.21±0.77 | 75.98±2.02 | 13.21±1.57 | 25.94±1.54 | 4.29±0.18 |
| | FLSD | 86.09±0.73 | 3.69±0.11 | 4.71±0.22 | 37.85±1.08 | 73.96±0.76 | 9.12±0.77 | 21.99±0.60 | 3.69±0.08 |
| | FL | 86.60±0.77 | 3.36±0.19 | 4.30±0.27 | 38.39±1.64 | 72.46±1.24 | 3.45±0.77 | 19.52±0.78 | 3.23±0.11 |
| | IFL | 85.85±0.29 | 2.92±0.16 | 3.97±0.13 | 41.19±1.40 | 72.31±0.57 | 11.21±0.32 | 24.46±0.49 | 6.31±0.07 |
| | DFL | 85.75±0.84 | 3.03±0.25 | 4.10±0.36 | **41.58±1.13** | 71.72±1.09 | **2.87±0.25** | 19.74±0.92 | **3.19±0.13** |
| | RO | 84.36±1.91 | 4.52±0.64 | 5.83±0.93 | 39.43±2.03 | 75.86±2.24 | 12.34±2.05 | 27.14±3.71 | 7.19±1.32 |
| | DRC | **87.46±1.56** | **2.83±0.43** | **3.66±0.62** | 37.96±1.62 | **71.18±2.54** | 6.39±1.35 | **19.34±2.38** | 3.52±0.41 |

| Dataset | Method | ECE (↓) | Brier (↓) | NLL (↓) | Dataset | Method | ECE (↓) | Brier (↓) | NLL (↓) |
|---|---|---|---|---|---|---|---|---|---|
| CIFAR -8-2 (Outlier) | ERM | 55.69±23.39 | 136.37±24.91 | 67.67±29.01 | CIFAR -80-20 (Outlier) | ERM | 64.26±0.39 | 151.27±0.57 | 106.94±1.80 |
| | PC | 70.60±10.18 | 151.27±13.78 | 99.21±42.13 | | PC | 64.50±1.19 | 152.05±1.46 | 95.59±2.37 |
| | LS | 36.33±11.02 | 110.98±9.95 | 27.55±2.13 | | LS | 29.65±0.92 | 116.05±0.73 | 54.92±0.62 |
| | FLSD | 57.10±0.44 | 134.02±0.63 | 50.15±0.73 | | FLSD | 45.59±0.39 | 129.95±0.48 | 76.85±0.56 |
| | FL | 67.07±0.65 | 146.57±0.99 | 66.37±0.74 | | FL | 53.16±0.35 | 138.38±0.37 | 86.54±0.65 |
| | IFL | 14.01±4.25 | 97.20±3.93 | 26.11±3.04 | | IFL | 46.29±0.38 | 133.23±0.50 | 79.58±0.86 |
| | DFL | 65.82±0.28 | 144.78±0.43 | 66.31±0.46 | | DFL | 52.10±0.26 | 137.08±0.34 | 86.46±0.28 |
| | RO | 74.56±1.08 | 156.72±1.46 | 79.05±2.43 | | RO | 65.14±2.38 | 152.38±2.96 | 96.58±8.82 |
| | DRC | **10.24±0.44** | **93.44±0.61** | **23.37±0.41** | | DRC | **15.38±0.85** | **109.05±0.49** | **49.97±0.35** |

Table 4: The comparison experimental results on different datasets and different methods under the strong augmentation setting. ↓ and ↑ indicate lower and higher values are better respectively. For better presentation, the best and second-best results are in **bold** and uderline respectively. For clarity, NLL values are multiplied by 10. Remaining values are reported as percentages (%). For datasets with both ID and outlier test samples, we report the results on all samples and outlier samples. We mark whether the test samples are sampled from ID or outliers in the table. *Compared with other methods,* DRC *achieves excellent performance on different metrics in almost all datasets.*

| Dataset | Method | ACC (↑) | EAURC (↓) | AURC (↓) | AUPR Err (↑) | FPR95% TPR(↓) | ECE (↓) | Brier (↓) | NLL (↓) |
|---|---|---|---|---|---|---|---|---|---|
| CIFAR-8-2 (All) | ERM | 79.35±0.27 | 0.70±0.02 | 2.99±0.07 | 83.77±0.83 | 16.29±1.22 | 6.89±0.78 | 25.86±0.77 | 6.78±0.31 |
| | PC | 79.65±0.17 | 0.64±0.03 | 2.87±0.01 | 84.54±1.18 | 12.80±1.53 | 4.94±0.44 | 24.35±0.29 | 6.66±0.20 |
| | LS | 79.65±0.10 | 4.57±1.80 | 6.80±1.78 | 75.48±3.08 | 29.96±5.26 | 10.01±0.40 | 28.05±0.89 | 7.62±0.34 |
| | FLSD | 78.22±0.18 | 2.33±0.06 | 4.89±0.05 | 70.65±0.83 | 50.43±0.94 | 8.58±0.36 | 32.14±0.14 | 10.54±0.11 |
| | FL | 78.96±0.09 | 1.07±0.19 | 3.46±0.20 | 78.62±1.84 | 31.57±6.61 | 9.51±1.84 | 29.34±1.73 | 8.78±1.28 |
| | IFL | 79.70±0.09 | 0.70±0.01 | 2.92±0.01 | 84.86±0.19 | 12.85±0.28 | 4.97±0.34 | 24.39±0.22 | 6.82±0.16 |
| | DFL | 78.69±0.25 | 1.97±0.13 | 4.42±0.07 | 71.32±1.06 | 48.43±2.15 | 13.25±0.24 | 33.70±0.05 | 12.47±0.38 |
| | RO | 75.38±4.42 | 2.03±0.47 | 5.48±1.78 | 76.19±3.88 | 43.03±2.23 | 17.36±1.22 | 40.15±4.98 | 23.41±3.95 |
| | DRC | **79.97±0.09** | **0.62±0.02** | **2.78±0.03** | 84.09±0.77 | **11.50±0.30** | **3.30±0.32** | **23.14±0.09** | **5.98±0.07** |
| CIFAR-80-20 (All) | ERM | 63.68±0.47 | 3.42±0.51 | 11.01±0.70 | 84.17±2.74 | 47.71±7.28 | 17.08±4.87 | 52.00±4.53 | 25.22±6.82 |
| | PC | 64.10±0.52 | 3.02±0.18 | 10.42±0.40 | 86.81±0.14 | 40.59±0.98 | 16.09±0.34 | 50.17±0.79 | 21.96±0.67 |
| | LS | 63.73±0.61 | 4.17±0.90 | 11.73±0.63 | 83.69±3.16 | 46.24±4.89 | **5.07±2.14** | 47.08±0.38 | 18.16±0.75 |
| | FLSD | 61.88±0.35 | 5.02±0.03 | 13.44±0.16 | 80.19±0.35 | 55.50±0.33 | 16.49±0.06 | 54.23±0.36 | 26.09±0.42 |
| | FL | 63.06±0.45 | 3.30±0.07 | 11.16±0.28 | 86.46±0.22 | 41.15±0.98 | 9.29±0.30 | 47.72±0.46 | 18.56±0.30 |
| | IFL | 64.43±0.62 | 3.02±0.09 | 10.27±0.33 | 86.66±0.48 | 41.42±1.10 | 21.66±1.07 | 54.25±1.46 | 29.84±1.70 |
| | DFL | 63.47±0.48 | 3.31±0.10 | 10.99±0.31 | 85.89±0.33 | 41.43±0.72 | 8.42±1.17 | 47.08±0.52 | 18.16±0.35 |
| | RO | 60.20±1.04 | 4.97±0.30 | 14.23±0.81 | 81.19±0.12 | 58.94±0.96 | 24.73±2.69 | 62.20±1.83 | 42.31±3.34 |
| | DRC | **65.90±0.33** | **2.37±0.11** | **8.98±0.24** | **88.38±0.10** | 34.76±0.29 | 9.84±0.40 | **44.59±0.44** | **17.20±0.19** |
| Image NetBG (All) | ERM | 87.35±0.14 | 0.98±0.02 | 1.82±0.04 | 61.85±1.00 | 44.06±1.25 | 3.47±0.10 | 17.79±0.30 | 4.04±0.07 |
| | PC | 86.88±0.36 | 1.09±0.06 | 1.99±0.11 | 61.90±0.71 | 45.12±2.37 | 7.19±0.46 | 19.95±0.89 | 5.21±0.35 |
| | LS | **87.54±0.39** | 1.26±0.02 | 2.07±0.07 | 60.55±0.61 | 44.85±0.89 | 4.42±0.27 | 17.73±0.47 | 4.20±0.12 |
| | FLSD | 86.64±0.34 | 1.21±0.04 | 2.15±0.09 | 61.42±0.94 | 47.22±0.32 | 7.51±0.38 | 19.56±0.33 | 4.50±0.07 |
| | FL | 86.70±0.20 | 1.17±0.07 | 2.10±0.08 | 62.08±1.79 | 46.23±2.16 | 3.92±0.13 | 18.65±0.37 | 4.21±0.09 |
| | IFL | 87.26±0.24 | 1.03±0.04 | 1.88±0.06 | 62.24±0.77 | 45.32±1.56 | 4.89±0.27 | 18.37±0.31 | 4.32±0.11 |
| | DFL | 87.11±0.48 | 1.17±0.06 | 2.04±0.13 | 60.91±1.27 | 46.51±0.79 | 2.96±0.26 | 18.11±0.66 | 4.09±0.15 |
| | RO | 87.21±0.29 | 1.23±0.05 | 2.08±0.08 | 59.35±1.32 | 48.24±1.78 | 6.22±0.26 | 19.22±0.33 | 4.76±0.09 |
| | DRC | 87.53±0.42 | **0.94±0.04** | **1.75±0.09** | 62.53±0.60 | 42.47±0.65 | **2.23±0.42** | **17.45±0.51** | **4.01±0.14** |
| Image NetBG (Outlier) | ERM | 83.52±0.15 | 1.68±0.03 | 3.12±0.06 | 62.37±1.06 | 53.02±1.13 | 4.61±0.10 | 23.13±0.37 | 5.25±0.10 |
| | PC | 82.91±0.47 | 1.83±0.11 | 3.38±0.19 | 62.65±0.57 | 54.36±2.34 | 9.40±0.63 | 25.93±1.17 | 6.77±0.46 |
| | LS | 83.81±0.52 | 2.07±0.03 | 3.46±0.12 | 61.05±0.67 | 53.05±0.90 | 4.21±0.30 | 22.86±0.62 | 5.31±0.16 |
| | FLSD | 82.63±0.43 | 2.05±0.08 | 3.66±0.16 | 61.96±0.92 | 54.95±0.86 | 7.92±0.44 | 24.99±0.44 | 5.67±0.10 |
| | FL | 82.72±0.25 | 1.99±0.11 | 3.57±0.14 | 62.59±1.83 | 54.38±2.00 | 4.04±0.17 | 24.01±0.48 | 5.38±0.12 |
| | IFL | 83.41±0.35 | 1.73±0.06 | 3.19±0.11 | 62.87±0.77 | 53.79±1.17 | 6.43±0.37 | 23.87±0.47 | 5.61±0.16 |
| | DFL | 83.28±0.64 | 1.97±0.10 | 3.46±0.21 | 61.44±1.35 | 54.90±0.88 | 3.03±0.35 | 23.36±0.86 | 5.24±0.21 |
| | RO | 83.41±0.37 | 2.04±0.09 | 3.51±0.14 | 59.99±1.34 | 56.03±1.06 | 8.14±0.34 | 24.93±0.41 | 6.18±0.12 |
| | DRC | **83.83±0.55** | **1.58±0.05** | **2.97±0.15** | **63.01±0.62** | 51.31±0.14 | 2.97±0.56 | **22.59±0.67** | 5.18±0.18 |
| Food 101 (ID) | ERM | 87.26±0.14 | 1.25±0.02 | 2.10±0.02 | 58.21±1.04 | 50.47±1.13 | 2.37±0.08 | 18.44±0.12 | 4.68±0.04 |
| | PC | 87.54±0.08 | 1.24±0.01 | 2.05±0.02 | 56.87±0.23 | 50.58±0.30 | 5.87±0.12 | 19.17±0.11 | 5.45±0.02 |
| | LS | 87.33±0.03 | 1.80±0.03 | 2.64±0.03 | 53.76±0.22 | 54.59±0.56 | 19.70±0.16 | 23.58±0.03 | 6.91±0.01 |
| | FLSD | 85.98±0.09 | 1.55±0.01 | 2.58±0.01 | 57.34±0.55 | 54.40±0.37 | 6.04±0.04 | 20.90±0.08 | 5.21±0.02 |
| | FL | 86.32±0.22 | 1.45±0.02 | 2.43±0.05 | 58.26±0.43 | 51.60±0.45 | 3.62±0.12 | 19.91±0.17 | 4.95±0.04 |
| | IFL | **87.67±0.07** | **1.22±0.04** | **2.01±0.03** | 57.87±0.80 | **49.71±1.12** | 5.05±0.07 | 18.58±0.07 | 5.10±0.01 |
| | DFL | 86.80±0.11 | 1.40±0.04 | 2.31±0.06 | 57.31±0.47 | 52.08±1.04 | **1.29±0.13** | 19.15±0.23 | 4.80±0.06 |
| | RO | 87.23±0.01 | 1.26±0.03 | 2.12±0.03 | 57.80±1.23 | 50.83±1.60 | 3.04±0.08 | 18.65±0.12 | 4.79±0.05 |
| | DRC | 87.32±0.12 | 1.25±0.01 | 2.09±0.04 | **58.27±0.42** | 50.33±0.76 | 2.17±0.13 | **18.34±0.13** | **4.67±0.02** |
| Came lyon (Outlier) | ERM | 90.21±0.38 | 1.41±0.06 | 1.91±0.10 | 38.69±0.55 | 63.63±0.88 | 4.60±0.19 | 14.89±0.51 | 2.59±0.08 |
| | PC | 87.30±0.40 | 1.97±0.10 | 2.81±0.14 | **40.47±0.75** | 70.33±0.54 | 9.91±0.80 | 21.92±1.14 | 5.09±0.54 |
| | LS | 92.52±0.48 | 1.28±0.16 | 1.57±0.19 | 34.84±0.71 | 58.96±1.14 | 18.69±0.36 | 18.66±0.47 | 3.49±0.06 |
| | FLSD | 92.24±0.28 | 1.12±0.05 | 1.43±0.07 | 35.29±0.31 | 59.69±1.19 | 12.81±0.27 | 15.83±0.17 | 2.85±0.03 |
| | FL | 92.30±0.24 | 1.11±0.04 | 1.41±0.06 | 35.17±0.12 | 59.67±1.09 | 12.91±0.28 | 15.82±0.10 | 2.85±0.02 |
| | IFL | 88.99±0.85 | 1.59±0.19 | 2.23±0.29 | 40.02±0.69 | 65.91±1.59 | 7.15±0.92 | 17.71±1.57 | 3.36±0.33 |
| | DFL | 90.60±1.30 | 1.72±0.21 | 2.19±0.34 | 36.44±2.74 | 61.35±0.52 | 7.18±0.96 | 15.39±0.93 | 2.66±0.11 |
| | RO | 91.55±0.58 | 2.62±0.33 | 2.99±0.38 | 31.57±1.27 | 64.19±2.47 | 7.10±0.60 | 15.28±1.16 | 5.31±0.58 |
| | DRC | **93.43±0.22** | **0.83±0.06** | **1.05±0.07** | 34.34±0.92 | **57.07±1.58** | 2.30±0.38 | **10.16±0.39** | **1.88±0.10** |

| Dataset | Method | ECE (↓) | Brier (↓) | NLL (↓) | Dataset | Method | ECE (↓) | Brier (↓) | NLL (↓) |
|---|---|---|---|---|---|---|---|---|---|
| CIFAR-8-2 (Outlier) | ERM | 26.42±2.31 | 103.12±1.77 | 27.70±1.06 | CIFAR-80-20 (Outlier) | ERM | 47.87±13.87 | 134.26±14.91 | 87.04±27.59 |
| | PC | 15.17±1.88 | 96.69±1.59 | 25.52±0.86 | | PC | 37.96±0.75 | 124.76±0.48 | 67.83±1.09 |
| | LS | 39.34±4.34 | 113.79±4.10 | 29.97±1.45 | | LS | 30.26±3.68 | 116.36±2.94 | 58.00±2.39 |
| | FLSD | 52.43±0.34 | 127.95±0.66 | 45.84±0.71 | | FLSD | 55.18±0.87 | 140.49±0.93 | 94.46±1.79 |
| | FL | 43.23±7.89 | 118.76±8.28 | 37.93±6.23 | | FL | 30.87±1.14 | 116.85±0.95 | 60.91±0.90 |
| | IFL | 14.40±1.39 | 96.78±0.89 | 25.34±0.60 | | IFL | 54.86±2.30 | 141.27±2.32 | 93.67±4.71 |
| | DFL | 60.51±1.02 | 138.24±1.58 | 55.67±2.37 | | DFL | 29.68±2.15 | 116.09±1.92 | 59.39±1.78 |
| | RO | 58.97±12.40 | 137.18±14.39 | 74.23±24.88 | | RO | 65.26±6.81 | 152.99±7.96 | 128.48±14.80 |
| | DRC | **8.81±1.51** | **92.69±0.66** | **22.93±0.35** | | DRC | **20.57±0.67** | **112.30±0.61** | **51.92±0.33** |

## C   PROOF OF THEOREM 1

Following Bai et al. (2021), we have

$$p - \mathbb{P}(Y = 1 \mid \hat{\mathbb{P}}_1(X) = p) = p - \mathbb{E}_Z[\sigma(\frac{\|\boldsymbol{w}^*\|}{\|\hat{\boldsymbol{w}}\|} \cos \hat{\theta} \cdot \sigma^{-1}(p)) + \sin \hat{\theta} \cdot \|\boldsymbol{w}^*\|Z],$$

where $\cos \hat{\theta} = \frac{\hat{\boldsymbol{w}}^\top \boldsymbol{w}^*}{\|\hat{\boldsymbol{w}}\| \cdot \|\boldsymbol{w}^*\|}$.

We first compute the calibration error for the baseline method

$$\hat{\boldsymbol{w}} = \arg \min_{\boldsymbol{w}} \frac{1}{n} \sum_{i=1}^n (\boldsymbol{w}^\top \boldsymbol{x}_i - \tilde{y}_i)^2,$$

where $\tilde{y}_i = (1 - \epsilon)y_i + \epsilon/2$. As $d/n = o(1)$, we have

$$\hat{\boldsymbol{w}} = (\mathbb{E}[x_i x_i^\top])^{-1}\mathbb{E}[x_i \tilde{y}_i] + o(1) = \frac{1 - \epsilon}{1 + \|\boldsymbol{w}^*\|^2}\mathbb{E}[x_i y_i] + o(1) = (1 - \epsilon)(1 - 2\eta) \cdot \boldsymbol{w}^* + o(1).$$

In the above derivation, the first equation uses Sherman–Morrison formula.

As a result, we have $\cos \hat{\theta} = 1 - o(1)$ as $n$ grows, and therefore when $n \to \infty$,

$$p - \mathbb{P}(Y = 1 \mid \hat{\mathbb{P}}_{baseline}(X) = p) = p - \sigma(\frac{1}{(1 - \epsilon)(1 - 2\eta)}\sigma^{-1}(p)).$$

Now, for the DRC, when the initialization parameter $\boldsymbol{\theta}^{(0)}$ satisfies $\|\boldsymbol{\theta}^{(0)} - \boldsymbol{w}^*\| \le c_1$ for a sufficiently small constant $c_1 > 0$, there will be only $o(1)$ outliers left, and therefore

$$p - \mathbb{P}(Y = 1 \mid \hat{\mathbb{P}}_{\text{DRC}}(X) = p) = p - \sigma(\frac{1}{1 - \eta}\sigma^{-1}(p)).$$

By the monotonicity of $\sigma$ and the nonnegativity of $\sigma^{-1}(p)$ when $p > 1/2$. We have

$$|p - \mathbb{P}(Y = 1 \mid \hat{\mathbb{P}}_{\text{DRC}}(X) = p)| < |p - \mathbb{P}(Y = 1 \mid \hat{\mathbb{P}}_{baseline}(X) = p)|.$$

Taking the expectation of $p$ for both sides, we have

$$ECE[\hat{\mathbb{P}}_{\text{DRC}}] < ECE[\hat{\mathbb{P}}_{baseline}].$$

