# OpenReview forum: "Provable Dynamic Regularization Calibration"
_ICLR.cc/2024/Conference — ICLR 2024 Conference Withdrawn Submission_

### Official Review · Reviewer_m76P · 2023-10-23

**Soundness:** 3 good
**Presentation:** 2 fair
**Contribution:** 2 fair
**Rating:** 3
**Confidence:** 4

**Summary:**

This paper addresses a problem that I believe to be of great significance: how best to produce a well-calibrated predictor.  While there are multiple angles from which to confront this issue (e.g. post-hoc recalibration), the authors focus on the approach of including a regularizer during training to address the well-observed issue that neural networks trained with cross entropy alone tend to be overconfident in their predictions.  Previous work along these lines has shown that such regularization can be helpful in mitigating this overconfidence.  However, these methods have worked by uniformly decreasing the confidence on all examples, despite the fact that there are some examples where high confidence is appropriate (while it is not for others).

The method proposed herein (DRC) works by identifying those examples for which high confidence is not appropriate throughout the training process, and applying a regularizer to only these examples.  This DRC technique is compared empirically to previous methods for achieving calibration through regularization, and shown to perform on par with or better than some existing methods.

**Strengths:**

-On a high level, this paper addresses an important issue that is steadily receiving more attention among deep learning practitioners.  If neural networks are to be reliably applied to problems in risk-sensitive domains like medicine, then calibration must be prioritized over 0/1 accuracy, since a patient with a 1% risk of disease must be treated differently than a patient with 40% risk, though a prediction of 0 is likely correct for both.

-With respect to the algorithm, I appreciate the approach taken in this paper.  Cross entropy training is not ideal for domains where calibration matters.  While post hoc recalibration may be applied after training, these methods require an additional dataset, and also limit the expressiveness of the original model.  Accordingly, it would be much better if calibration were induced by the training process so that neural network predictions may have reliable confidence estimates out of the box.  I think the idea of instance-wise regularization is a step in the right direction for training well-calibrated networks.

-I appreciate the effort to rigorously motivate the approach by including the analysis of the theoretical construction.

**Weaknesses:**

-I think there is a significant gap in the discussion of related work (and thus the empirical evaluation).  Besides regularization-based techniques, there has been progress in improving calibration in the training process by using auxiliary loss functions that directly measure calibration error.  For example, in well-cited work Kumar et al (2018) proposed the MMCE kernel-based loss function and showed it to be very effective at reducing calibration error during training.  Additionally, Karandikar et al (2021) propose soft calibration objectives S-AvUC and SB-ECE as secondary losses which optimize for calibration throughout training.  It does not seem possible to motivate the DRC algorithm without at least mentioning this work, and explaining why it is not necessary for comparison.  Another clear oversight in my opinion is with respect to Mix-up training.  Mix-up training has been shown to improve calibration, and in my experience and research I have found it to be an improvement over generic label smoothing for the purpose of calibration.  I think that MMCE, S-AvUC, SB-ECE, and mixup would need to be included both in the related work as well as the empirical evaluation in order for this work to be sufficiently thorough that conclusions can be drawn.

-The explanation of the method seems to take for granted that outliers should be automatically assigned lower confidence.  However, this seems to be a heuristic that will not always hold.  For example, a dataset may be primarily composed of “difficult” examples, and then contain a smaller subset of “easier” examples further from the classification boundary.  In this case, it would seem that such a method would increase/decrease confidence on the wrong examples.

-Because of the two issues mentioned above, the paper is not clear or convincing in its motivation.  Since there has been significant research with regard to non-regularization based approaches to training for better calibration, in order to set these methods aside there needs to be further discussion as to why regularization based approaches are preferable, or at least why they should be considered separately.  In addition, I find the discussion of over-/under-confidence, easy/hard examples, outliers, and known/unknown knowns to be lacking rigor, and thus unconvincing.  As these arguments seem to form the basis of the motivation for the algorithm, they would need to be made more precise and understandable in order to achieve their aims.

-Besides the omission of important baselines, I find the empirical study to be difficult to draw conclusions from.  First, I am not sure where the weak vs. strong augmentation question comes into play.  It seems this technique is proposed for (and could be applied to) the typical NN training setup, and augmentation is barely mentioned before the experiments section, so it is confusing why it is featured so heavily there.  Also, the lack of information in general regarding hyperparameters (both for DRC and comparison techniques) creates concerns about the reproducibility of the experiments and competitiveness of baseline implementations.

-The theory section is not well-motivated or general enough to lead to any useful conclusions.  Squared loss is a particularly unusual choice here, since CE is used everywhere else, and the choice of label smoothing and the data generating model are not well-motivated.

**Questions:**

While I think this paper takes an interesting approach to confronting an important and open challenge, I think the current version is quite flawed.  The motivation seems to rely on heuristics that are never made rigorous, and the lack of clarity and omission of key comparisons in both the motivation and empirical evaluation make it difficult to judge its merits and see any significant value in it currently.

Clarifications on the below issues would be helpful for my understanding:

-How were the set of comparison methods chosen?  Why is it appropriate to only consider regularization techniques (as opposed to e.g. MMCE), and why was Mixup not included?

-Can you further explain the use of augmentation in the experiments?

-What hyperparameters were used for each baseline, and how were they chosen?

-Why were the particular data generating model and baseline model chosen for the theoretical analysis?  Is there a model besides label smoothing that would lead to different conclusions?  What about ERM, as that would be the most obvious comparison?

**Details Of Ethics Concerns:**

None.

---

### Official Review · Reviewer_wg6U · 2023-10-30

**Soundness:** 2 fair
**Presentation:** 2 fair
**Contribution:** 2 fair
**Rating:** 3
**Confidence:** 3

**Summary:**

This paper proposes dynamic regularization calibration (DRC) to make the model more robust and calibrated. The paper provided comprehensive experiment results and some theoretical justification.

**Strengths:**

1. There are comprehensive experiment results presented in this paper.
2. This paper discusses the intuition well, making the paper easy to understand and easy to follow.

**Weaknesses:**

1. It would be better to make the introduction and abstract more clear, instead of mainly using intuition. Some claims lack justifications. For example, "In summary, the lack of clear supervision on how to adjust confidence may lead to three significant
problems of existing methods..." The paragraph is not supported well.
2. This paper mainly addresses the issue when there are outlier data, not addressing the general model mis-calibration issue. Even if all the data are from "in distribution", DL models will still suffer from mis-calibration. The paper should've reflected this clearly in the title / abstract.
3. The assumptions are not very reasonable. I am not fully convinced that model mis-calibration are mainly due to outlier data. However, the paper assumes that training set drawn from distribution P follows Huber’s η-contamination model. It would be better if the authors can discuss in more detail why this assumption is reasonable.
4. The theory presented in this paper is not very convincing either. First, it assumes Huber’s η-contamination model. Second, the theory contains some technical conditions that are hard to interpret, e.g. We assume ∥w∗∥ ≤ c0 for some sufficiently small c0 > 0. It would be better to discuss why these assumptions are reasonable.

**Questions:**

Same as Weakness Section.

---

### Official Review · Reviewer_nfky · 2023-10-31

**Soundness:** 3 good
**Presentation:** 3 good
**Contribution:** 3 good
**Rating:** 5
**Confidence:** 4

**Summary:**

This paper proposes a training-time calibration scheme in which the loss for a given sample in a training batch is either set to the standard negative log-likelihood or the cross-entropy with a uniform distribution, depending on where the negative log-likelihood for that sample ranks relative to other samples in the same batch (effectively a style of per-batch outlier detection). The proposed approach is shown to be empirically powerful, outperforming various baselines in terms of both accuracy and ECE across multiple datasets.

**Strengths:**

- **Originality:** There are a number of sample-dependent adaptive calibration approaches in the literature, but the particular approach in this paper seems new to me.
- **Quality:** The algorithm described in the paper is straightforward to understand and easy to implement, and the authors demonstrate that it has strong empirical performance with this performance being accentuated in the presence of outliers.
- **Clarity:** Overall the paper is relatively easy to follow, although some aspects highlighted under weaknesses below could be made clearer.
- **Significance:** The results in the paper are impressive, and it seems significant that the intuition of outlier detection behind the proposed algorithm actually translates to practice on datasets with outliers.

**Weaknesses:**

## Main Weaknesses
1. **Theory is unclear/does not add anything.** I personally think Section 5 in the paper could be replaced entirely with further experiments to improve the paper. Mainly, my issue is that considering linear regression on label-smoothed data and then using the fit model as a logistic regression is not something anyone would do in practice. Additionally, even with this model, Theorem 1 assumes that we are basically initialized at an optimum. Also, the proof logic regarding DRC doesn't make sense to me (or rather, what is DRC even supposed to mean in this context given that it was originally defined in the batch optimization setting)? Lastly, what is $k = 2, ..., K$ doing in the statement of Theorem 1? This is the binary classification setting.

2. **Empirical claims should have further clarification.** The empirical results in Section 4 are impressive, but I think they could be improved with additional information. For example, what was the hyperparameter tuning strategy for the compared-to methods? Section B.6 goes into detail about tuning for the proposed method, but I could not find similar information for the other methods. Additionally, it would be useful to mention computational costs - it is a priori not obvious that all of the methods considered take comparable amounts of (wall clock) time to train, due to the per-batch overhead from proposed method (sorting and then selectively using the two different losses). Lastly, error bounds for the reported metrics (i.e. 1 std after averaging over several runs) would be useful in Table 1; they seem to be reported in the tables in the appendix (although the $\pm$ quantities are not explained even in these tables - what are they?).

3. **Missing coverage of related work/ideas.** Data augmentation techniques such as Mixup [1, 2] should also be mentioned in the related work discussion alongside focal loss and label smoothing, and if possible compared to in the experimental results since they are also very commonly used for improving calibration in a data-dependent way. Furthermore, a related strategy to the approach taken in this paper is explored in the recent work of [3], which is not mentioned.

## Minor Comments
- The motivating paragraph towards the end of the introduction is a bit tricky to read ("known knowns" and "known unknowns"); while I understand what the authors are trying to say here, I think it would be clearer to specify what is meant as a "known sample" (i.e. an easy-to-classify sample) at the beginning of the paragraph.
- The definition of $p_u$ should be re-introduced (i.e. mention it's uniform distribution) in the discussion of the DRC loss at the end of section 3.
- The model backbones should be mentioned in the main paper in some capacity; as it stands, there are no references to what models were actually trained in the main paper.
- It would (personally) be a little clearer to use the (Out.) and (Full.) qualifiers over every metric name so there is no possibility of getting confused in Table 1.

## Recommendation
I am a little worried about the theoretical component in this paper, as it makes little sense to me in its current form. That being said, the experimental results in the paper are impressive - with a little further clarification regarding how tuning was done for the other approaches, I lean accept for this paper. For now I recommend **weak reject** awaiting responses from the authors regarding the questions stated above (and also below); I am happy to update my rating accordingly after discussion.

[1] https://arxiv.org/abs/1710.09412
[2] https://arxiv.org/abs/1905.11001
[3] https://aclanthology.org/2022.emnlp-main.664.pdf

**Questions:**

My main questions are detailed under weaknesses above, but some additional questions include:

- Can you provide greater intuition regarding the proposed approach? My understanding is that we are actually regularizing the examples with *high* loss, which are the ones in which our predicted probabilities for the correct class are actually already not spiky. This is in direct contrast with approaches like focal loss; is the idea here that we should effectively "give up" on outliers?
- Was using a burn-in period for DRC ever considered? I.e. train regularly for an epoch or two and then switch to DRC? This intuitively seems like it would better allow for relative difficulty to be ascertained.

---

### Official Review · Reviewer_Vf1p · 2023-11-04

**Soundness:** 2 fair
**Presentation:** 1 poor
**Contribution:** 1 poor
**Rating:** 3
**Confidence:** 4

**Summary:**

The paper proposes a regularization technique for supervised neural networks that improves uncertainty estimation. The technique assumes that the dataset is split into easy and hard examples and maximizes confidence on easy examples while maximizing entropy on the hard examples.

**Strengths:**

- The paper studies an important problem, uncertainty estimation in neural network
- The method is simple and easy-to-use

**Weaknesses:**

- From the point of view of clarity, the paper is understandable, but not sufficiently polished in my opinion. The high-level motivation centers around the point that existing methods simultaneously optimize for confidence and entropy, but this point is not made crisply. The authors make a very general statement that be made about any regularization method (e.g., L2 regularization penalizes weight magnitude and conflicts the goal of training accuracy...). I would encourage the authors to better motivate their method in future versions of this paper. Also, the writing itself is not polished, with many grammatical errors or missing words.
- The assumption that the data can be split into easy and hard examples is strong and not well-justified. The paper should have included a discussion on why that assumption in easy to make and what are some possible strategies of splitting the data so that the method can be applied. The trick used in the experiments of splitting CIFAR10 into different classes is artificial.
- The experimental results are not convincing: while the method does appear to produce better metrics, the difference relative to the baselines is small, and since none of the metrics have error bars, it's not possible to tell whether there is actually any improvement.
- Lastly, I think the paper is missing an important baseline: post-hoc confidence recalibration (e.g., Platt, 1999 for classification problems; Kuleshov et al., 2018 for regression problems). It's unclear to me why this regularization approach would be preferable to post-hoc recalibration and why it would work better.

**Questions:**

- Why would the method work better than post-hoc recalibration?
- How would one construct real-world "hard datapoints" in practice?